# Stability and nuclear localization of yeast telomerase depend on protein components of RNase P/MRP

P. Daniela Garcia[1], Robert W. Leach [2], Gable M. Wadsworth[3], Krishna Choudhary[4,5], Hua Li[4], Sharon Aviran [4], Harold D. Kim [3] & Virginia A. Zakian[1✉]

RNase P and MRP are highly conserved, multi-protein/RNA complexes with essential roles in processing ribosomal and tRNAs. Three proteins found in both complexes, Pop1, Pop6, and Pop7 are also telomerase-associated. Here, we determine how temperature sensitive *POP1* and *POP6* alleles affect yeast telomerase. At permissive temperatures, mutant Pop1/6 have little or no effect on cell growth, global protein levels, the abundance of Est1 and Est2 (telomerase proteins), and the processing of TLC1 (telomerase RNA). However, in *pop* mutants, TLC1 is more abundant, telomeres are short, and TLC1 accumulates in the cytoplasm. Although Est1/2 binding to TLC1 occurs at normal levels, Est1 (and hence Est3) binding is highly unstable. We propose that Pop-mediated stabilization of Est1 binding to TLC1 is a pre-requisite for formation and nuclear localization of the telomerase holoenzyme. Furthermore, Pop proteins affect TLC1 and the RNA subunits of RNase P/MRP in very different ways.

[1] Department of Molecular Biology, Princeton University, Princeton, NJ 08544, USA. [2] Bioinformatics Group, Genomics Core Facility, Carl Icahn Laboratory, Princeton University, Princeton, New Jersey 08544, USA. [3] School of Physics, Georgia Institute of Technology, Atlanta, Georgia 30332, USA. [4] Department of Biomedical Engineering and Genome Center, University of California, Davis, California 95616, USA. [5] Gladstone Institute of Data Science and Biotechnology, San Francisco, CA 94158, USA. ✉email: vzakian@princeton.edu

Telomerase is a ribonucleoprotein complex whose RNA component is the template for extending the G-rich strand of telomeric DNA. As in other organisms, *Saccharomyces cerevisiae* telomerase consists of both an RNA and multiple protein subunits (reviewed in ref. [1]). The RNA component, TLC1, is a large molecule (~1200 nucleotides) with a complex secondary structure. Multiple proteins are TLC1-associated including the three Est proteins, Est1, Est2, and Est3, the heterodimeric Yku complex and the ring-shaped heptameric Sm (Sm$_7$) complex (Fig. 1a). TLC1 and the three Est proteins are essential for telomerase action in vivo[1]. Est1 is the only telomerase subunit whose abundance and activity are cell cycle regulated, peaking in late S/G2 phase[2–6]. As in most organisms, yeast telomerase is not abundant: haploid cells contain ~40–80 molecules of the Est proteins[4,7] and ~30 molecules of TLC1[8].

Biogenesis of TLC1 is complex as it undergoes several processing and intracellular trafficking events[1] (Fig. 1b). TLC1 is transcribed by RNA polymerase II to make a ~1300 nt transcript[9,10]. The TLC1 transcript has a 7-methyl-guanosine (m7G) cap at its 5′ end[11]. TLC1 can acquire a 3′ polyadenylated [poly(A)] tail, although the active form of TLC1 lacks poly(A)[12]. TLC1 then transits to the nucleolus where the 5′ m7G cap is hypermethylated[11,13]. Next TLC1 moves to the cytoplasm where the Est proteins bind[13]. Telomerase returns to the nucleus to elongate telomeres[13,14] (Fig. 1b). If TLC1 is unable to exit the nucleus, as occurs when its export factors are missing, assembly of telomerase is blocked and telomere length is compromised[15].

There are two pathways that recruit different forms of telomerase to the nucleus. Est2, the reverse transcriptase, binds the central core of TLC1 (Fig. 1a). The Yku heterodimer binds to a stem-loop region at the end of one of the structured arms of TLC1[3,16]. This TLC1-Yku interaction is required for TLC1-Est2 to enter the nucleus where it binds telomeric chromatin in G1 phase by its association with telomere-associated Sir4[3,13,16–18]. However, this G1 telomerase is not competent to elongate telomeres as it lacks Est1 and Est3, and the complex is not associated with the G-tail[19]. Est1, which binds directly to both TLC1 and Est3 (Fig. 1a), has two important functions. Est1 is required for the second recruitment pathway, which occurs in late S/G2 phase[20] via a direct interaction between Est1 and telomere-bound Cdc13[2,7,21]. Est1 also activates telomerase in late S/G2 phase, perhaps by bringing Est3 to telomeres[2–4,18]. Although the Est1 activation step is essential for telomerase action, either recruitment pathway is sufficient to maintain stable (albeit short) telomeres[18].

Mutations in ~400 genes affect telomere length, although in many cases, the mechanism by which they do so is unknown. To identify previously unknown regulators of telomerase, we carried out mass spectrometry (MS) to identify proteins that are telomerase-associated in vivo[5]. Using this method, we identified 115 high confidence telomerase-bound proteins. About 35% of the telomerase-associated proteins function in RNA biogenesis, including Pop1, Pop6, and Pop7. The three Pop proteins were equally telomerase-associated in G1 and late S/G2 phase, suggesting that they are associated with telomerase throughout the cell cycle. Our demonstration that the three Pop proteins associate with telomerase[5] was confirmed by a subsequent MS study[22].

Pop1, 6 and 7 are components of the highly conserved RNase P and RNase MRP multi-protein-RNA complexes[23]. The RNase P complex processes tRNAs by cleaving a 5′ precursor to produce mature tRNA. RNase MRP processes mitochondrial RNA, the 35s rRNA precursor, and mitotic cyclin B2 mRNA. Although most of the proteins that make up RNase P/MRP are found in both complexes, each complex contains a different RNA. The RNA subunits of RNase P and RNase MRP are, respectively, the 369 nt RPR1 and the 340 nt NME1. The genes encoding the two RNAs and all of the protein subunits of RNase P/MRP are essential. The Pop proteins are thought to promote the proper folding of RPR1 and NME1 and to mediate substrate recognition by the two complexes[23]. Of the eleven proteins in RNase P/MRP, only three are telomerase-associated with high confidence[5]. Pop proteins are found in the three cellular compartments through which TLC1 transits: nucleoplasm, nucleolus, and cytoplasm[23]. Therefore, Pop proteins could associate with telomerase at any (or all) of these locations.

Like TLC1, RPR1 and NME1 are highly structured RNAs. The Pop6/Pop7 heterodimer binds a stem-loop region called the P3 domain in RPR1 and NME1, which then stabilizes the subsequent binding of Pop1 to the P3 region[23,24]. Pop1 is thought to affect the global folding of RPR1 and NME1 in part by recruiting other proteins to the complex[24]. TLC1 also has a P3-like region, the CS2a/TeSS domain, that is also bound by Pop6/Pop7[22] (Fig. 1a). Remarkably, the P3 regions of NME1, RPR1, and TLC1 are functionally interchangeable[22,25]. Although the Est1 and Pop binding sites are separable (Fig. 1a), deletion of the CS2a/TeSS domain was reported to prevent Est1 binding to the mutant TLC1[22]. Addition of Pop1 enhances telomerase activity in vitro. However, the mechanism by which Pop proteins affect telomerase in vivo is not known nor is it known if Pop proteins affect telomerase abundance or telomere length.

Here, we used an in vivo approach to determine how Pop1 and Pop6 affect telomerase and telomeres using *pop* temperature-sensitive alleles. At the permissive temperature of 24 °C, *pop* cells grow normally and have near wild type (WT) levels of protein, including Est1 and Est2. However, telomeres are ~30% shorter in Pop deficient cells, and telomerase binding to telomeres is low. The *pop* cells have twice as much TLC1 compared to WT cells, while NME1 levels are decreased. The global secondary structure of NME1, but not TLC1, is perturbed in *pop* mutants. Thus, even though Pop1 and 6 bind to similar structures in TLC1 and NME1[22], they affect the two RNAs in opposite ways. In contrast to an earlier study[22], DMS protection showed that levels of Est1 and Est2 binding to TLC1 are normal in *pop* cells. However, this binding is highly unstable. Moreover, TLC1 accumulates in the cytoplasm in *pop* cells, even though *pop* cells have normal levels of Est proteins and TLC1 export factors. We propose that Pop1, 6, and 7 stabilize the cytoplasmic association of Est1 (and to a lesser extent Est2) with TLC1, and the failure to do so prevents formation of a stable holoenzyme, which traps TLC1 in the cytoplasm.

## Results

**Strains to study effects of reduced Pop proteins**. The goal of this paper is to determine the functional significance of the association of Pop proteins with telomerase. This goal is complicated by the fact that all of the subunits of RNase P/MRP are essential to process tRNAs and ribosomal RNAs. Therefore, the absence of any one subunit ultimately stops protein synthesis. Direct effects of Pop proteins on telomerase must be distinguished from their effects on protein synthesis.

Experiments were carried out using temperature sensitive *pop* strains and their isogenic WT controls[26]. Although the *pop* strains were known to be temperature sensitive, they were otherwise uncharacterized. While some experiments were conducted in *pop7-PH* cells, the short telomere phenotype in this strain was not completely suppressed by introducing a plasmid-borne *POP7* gene. Therefore, we report only experiments carried out in *pop1-500* and *pop6-502* (hereafter, *pop1* and *pop6*) cells. Sequencing the mutant *pop1* revealed that it had seven amino acid changes spread throughout the 875 amino acid protein (Fig. 2a). *POP6* encodes a 158 amino acid protein. The mutant

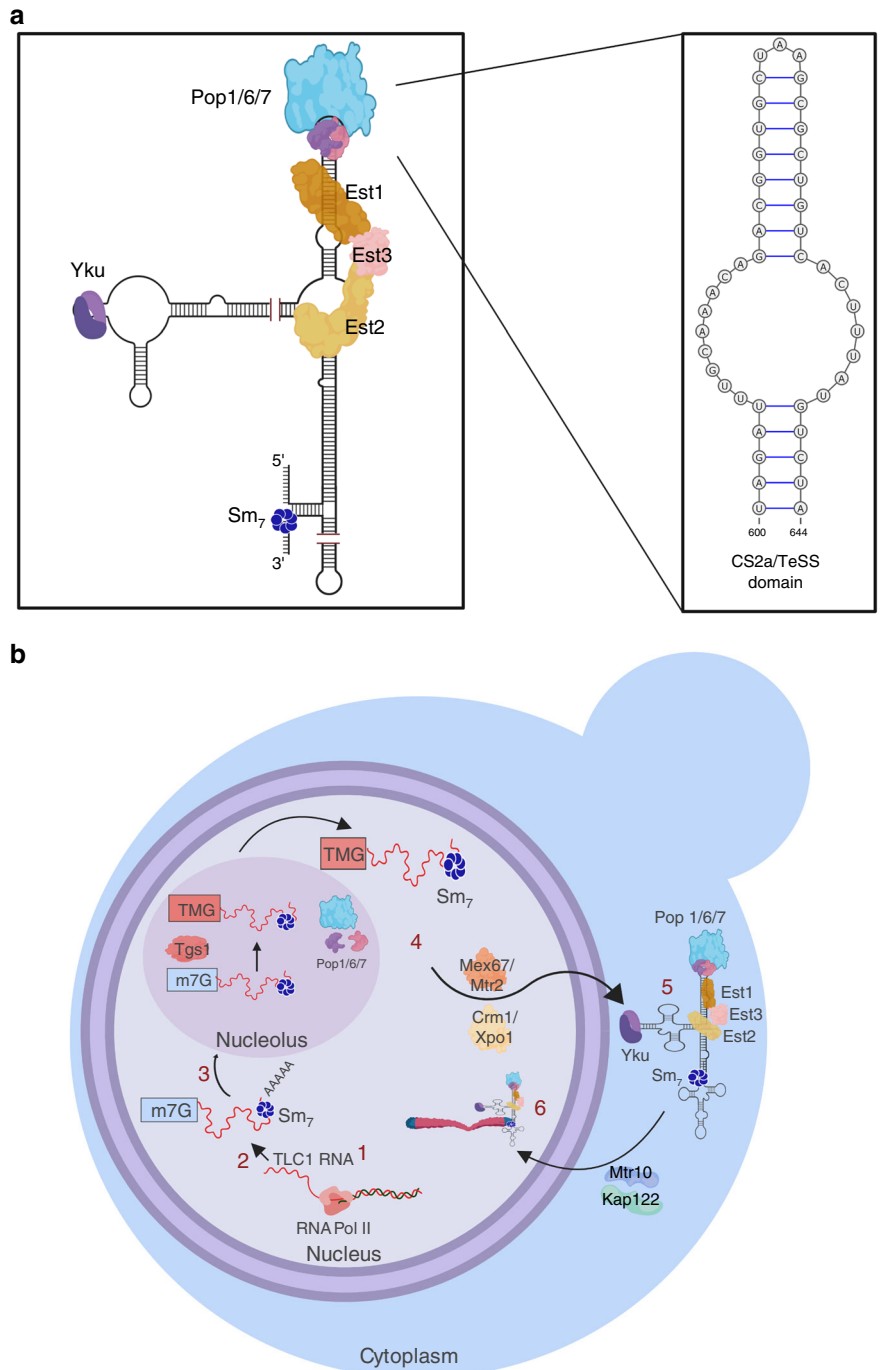

**Fig. 1 Structure and biogenesis of TLC1. a** Est1 and Pop proteins bind at separable sites near the end of the Est1 arm of TLC1. Est3 interacts directly with Est1 and Est2, possibly bridging the two, and both of these associations are required for Est3 to bind telomeres. Est2 binds the central core of TLC1. (The proteins and RNA are not drawn to scale; 1a is a static representation meant to illustrate the sites on TLC1 to which the indicated proteins bind and the protein-protein interactions amongst the telomerase subunits.) The binding sites for the heterodimeric Ku complex and the Sm7 complex are also shown. Insert shows magnified view of the CS2a/TeSS domain to which a Pop6/7 heterodimer binds and then recruits Pop1[22]. **b** Biogenesis of TLC1: (1) TLC1 is transcribed in the nucleus by RNA polymerase II. (2) The newly transcribed TLC1 has a 5′-7 methylguanosine cap, is bound by the Sm7 complex which helps stabilize the RNA[11] and a fraction of molecules have a poly(A) 3′tail. (3) TLC1 transits to the nucleolus where the 5′ cap gets hypermethylated by the Tgs1 methyltransferase. (4) TLC1 is bound by the indicated export factors that bring it to the cytoplasm. (5) TLC1 lacking a poly(A) tail assembles with the Est proteins in the cytoplasm. (6) In G1 phase, when Est1 abundance is low, Est1 and Est3 are not TLC1-associated. However, a Yku-TLC1-Est2 complex forms and is telomere associated in G1 phase. In late S/G2 phase, the holoenzyme forms in the cytoplasm and binds import factors Mtr10/Kap122 that mediate holoenzyme entry into the nucleus. The holoenzyme binds and elongates telomeres. Pop proteins are present in the nucleoplasm, nucleolus, and cytoplasm. The compartment in which Pop proteins bind TLC1 is not known. However, Pop proteins are TLC1-associated in both G1 and G2/M phase (see text for references). Images were made in BioRender (biorender.com).

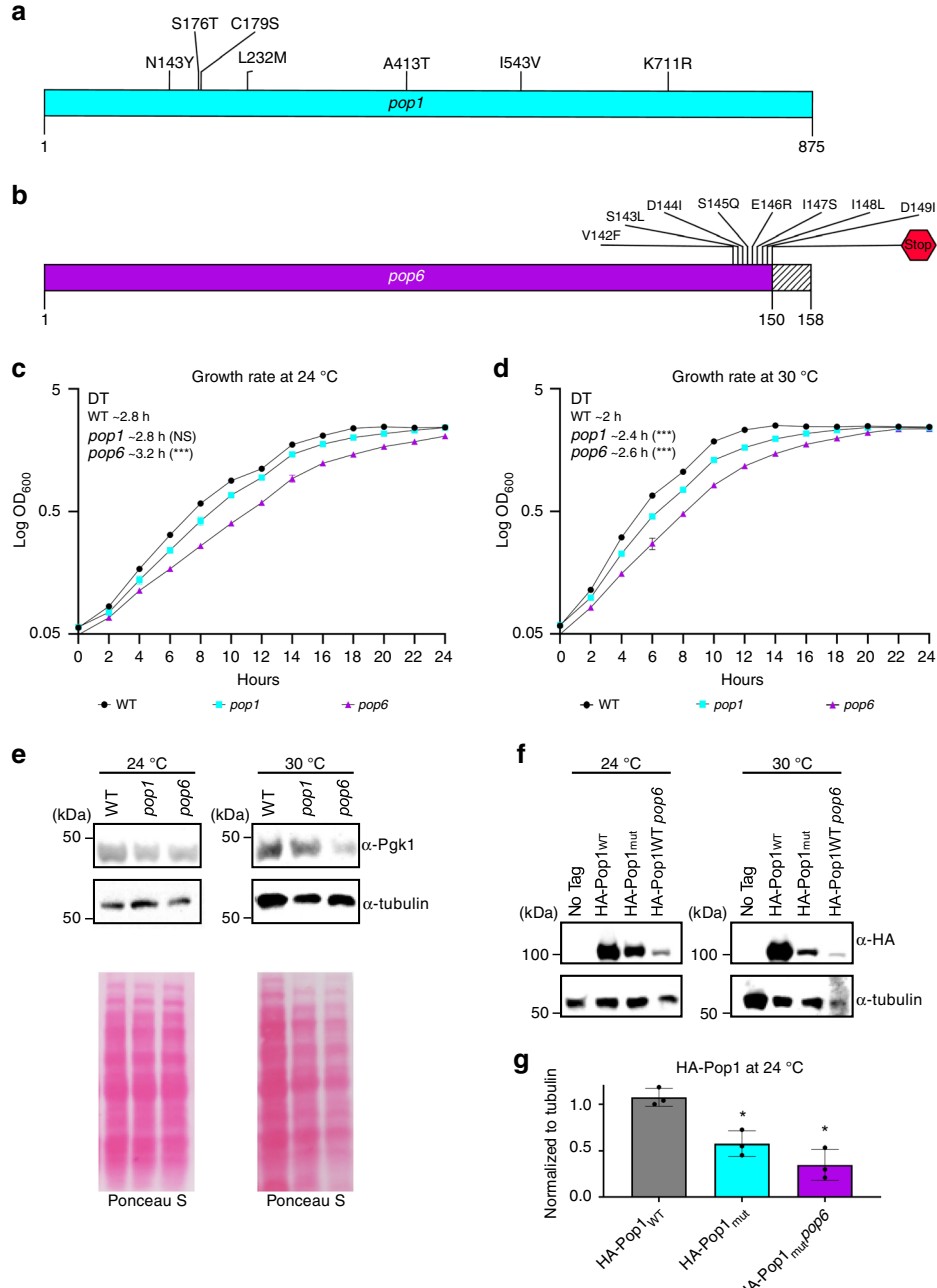

**Fig. 2 Growth and global protein levels are similar in *pop* and WT cells at 24 °C.** Diagrams of the proteins encoded by the *pop* alleles used herein. Silent mutations are not shown. **a** The positions of the seven amino acid substitutions in the 875 amino acid protein encoded by the *pop1* allele (blue) are shown. The nucleotide mutations that led to these substitutions are: 225 G → A, 427 A → T, 526 T → A, 535 T → A, 695 A → T, 792 T → C, 1237 G → A, 1627 A → G, 2132 A → G (numbers are positions of altered nucleotides). **b** Diagram of the protein produced by the *pop6* allele (purple), which contains a deletion at nucleotide 421 resulting in an eight amino acid substitution (amino acids 142–149) and loss of the nine terminal aminoacids (dashed lines from 150–158) due to a stop codon at position 150. Growth rates of *pop1* and *pop6* cells at **c** 24 °C and **d** 30 °C. **e** Pgk1 and tubulin proteins from WT, *pop1* or *pop6* cells grown for ~50 generations were analyzed by western blotting. Although in this western, there appears to be more Pgk1 in WT than in *pop* cells at 24 °C, using a two-tailed Student's *t*-test, the differences were not significant. Total protein was stained with Ponceau S. **f** Western blots of proteins from a no tag control strain or WT or *pop6* cells expressing HA-Pop1$_{WT}$ or cells expressing HA-Pop1$_{mut}$. **g** Quantification of HA-Pop1 levels from biological triplicates (black circles) of WT (gray bar), *pop1* (blue bar) and *pop6* (purple bar) grown at 24 °C were normalized to tubulin and WT protein levels. (Because, epitope tagged mutant Pop6 did not support viability, the effects of temperature on Pop6 were not studied). Error bars are one standard deviation from the average value of three or more independent experiments. *P*-values were calculated using unpaired two-tailed Student's *t*-test; *$P \leq 0.05$, **$P \leq 0.01$, ***$P \leq 0.001$, ****$P \leq 0.0001$; NS, not significant, $P > 0.05$. Source data are provided as a Source Data file.

*pop6* allele had a single nucleotide deletion near the end of the open reading frame. This deletion changed the sequence of 8 amino acids, as well as introducing a stop codon that resulted in a nine amino acid truncation (Fig. 2b).

To establish the appropriate conditions for experiments, we determined the growth rates of *pop* mutants at different temperatures. At 24 °C, *pop1* and *pop6* cells had doubling times similar to that of WT in both liquid and solid media (Fig. 2c;

Supplementary Fig. 1). At 30 °C, *pop1* and *pop6* cells grew indefinitely, but their growth rates were significantly slower than WT (Fig. 2d; Supplementary Fig. 1). At 37 °C, both mutants divided only a few times. Thus, 24 °C, 30 °C and 37 °C are, respectively, permissive, semi-permissive, and non-permissive temperatures for *pop1* and *pop6* cells.

Global protein levels were similar in *pop1*, *pop6* and WT cells at 24 °C, but were detectably lower at 30 °C (Fig. 2e). By western blot, alpha tubulin and Pgk1 protein levels were similar to WT at 24 °C but only 0.5× WT at 30 °C (Fig. 2e). Levels of mutant Pop1 (HA-Pop1$_{mut}$) were reduced even at 24 °C (0.6× WT Pop1). In addition, at 24 °C, HA-Pop1$_{WT}$ was less abundant in *pop6* than in WT cells (0.3× WT levels; Fig. 2f, g). Therefore, the abundance of some (and perhaps most) proteins was largely unaffected in *pop*

cells at 24 °C. However, mutant Pop1 and probably Pop6 were unstable even at permissive temperatures. Based on these results, all experiments were carried out at 24 °C as growth rates and bulk protein levels were similar to WT, but Pop1 (and probably Pop6) levels were reduced. Many experiments were also conducted at 30 °C. We reasoned that a phenotype that was detected at permissive temperature (24 °C) and worsened at semi-permissive temperature (30 °C) was likely a direct effect of Pop proteins on telomerase.

**pop mutants have shorter telomeres than WT cells.** To determine if Pop proteins are needed for WT telomere length, telomeres were measured by Southern blot analysis (Fig. 3). The

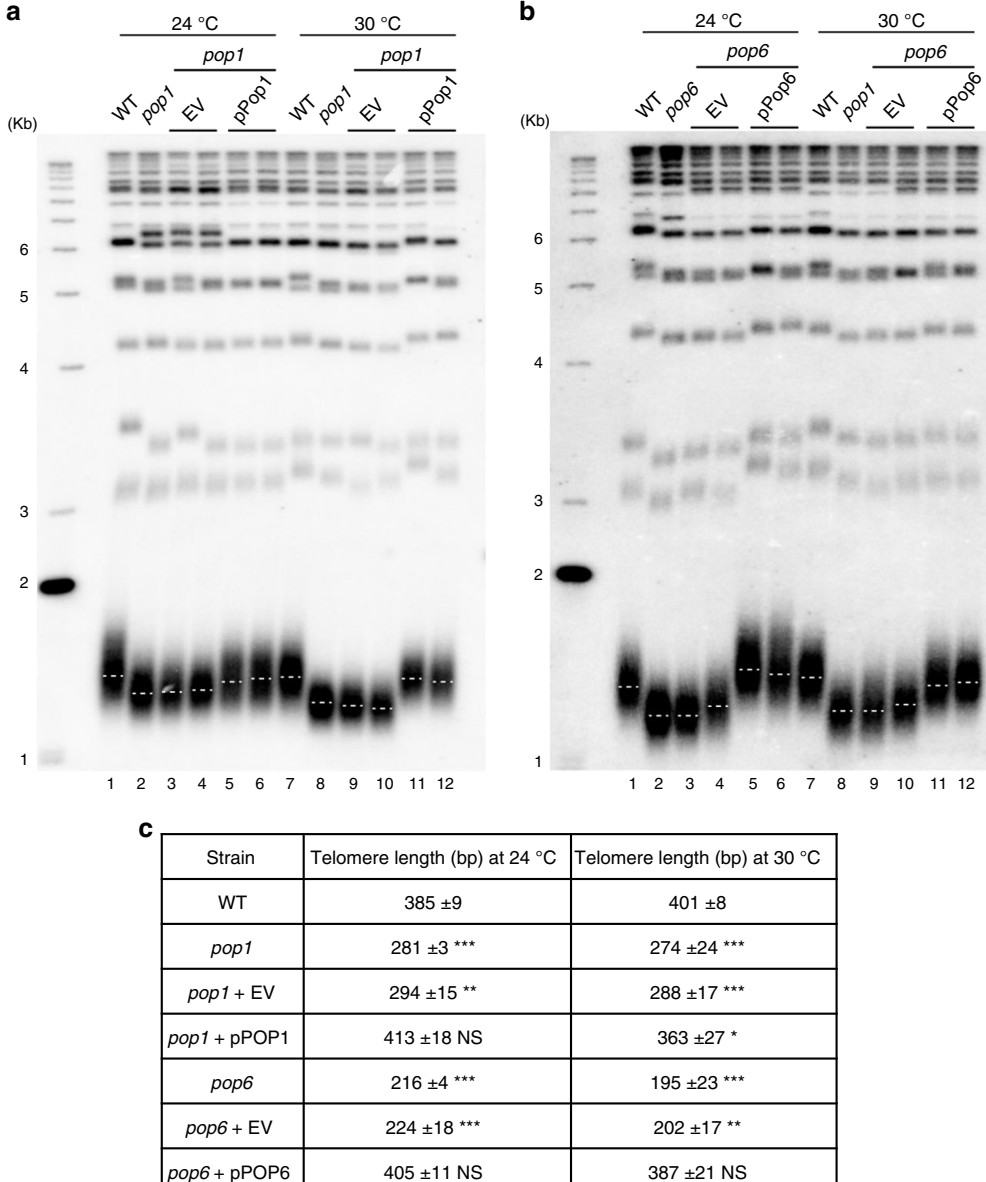

| Strain | Telomere length (bp) at 24 °C | Telomere length (bp) at 30 °C |
|---|---|---|
| WT | 385 ±9 | 401 ±8 |
| *pop1* | 281 ±3 *** | 274 ±24 *** |
| *pop1* + EV | 294 ±15 ** | 288 ±17 *** |
| *pop1* + pPOP1 | 413 ±18 NS | 363 ±27 * |
| *pop6* | 216 ±4 *** | 195 ±23 *** |
| *pop6* + EV | 224 ±18 *** | 202 ±17 ** |
| *pop6* + pPOP6 | 405 ±11 NS | 387 ±21 NS |

**Fig. 3 pop1 and pop6 cells have short telomeres at 24 and 30 °C that is reversed by introducing a WT copy of POP1 or POP6.** DNA was isolated from the indicated strains, digested with XhoI, and analyzed by Southern blotting using a radio-labeled TG$_{1-3}$ probe. Cells were grown at 24 or 30 °C for ~75 generations prior to DNA extraction. **a, b** EV, empty vector; pPOP1 or pPOP6, the same vector containing POP1 or POP6. Dotted white lines mark the average telomere length in each sample. **c** Quantitation of telomere lengths in at least three biological isolates of each strain. Size of telomeric tracts were determined by subtracting 875 nts of Y' DNA from the sizes of the terminal XhoI fragments. Introducing a WT copy of the POP gene reversed telomere shortening in *pop1* and *pop6* cells. P-values were calculated using unpaired two-tailed Student's t-test; *P ≤ 0.05, **P ≤ 0.01, ***P ≤ 0.001, ****P ≤ 0.0001; NS, not significant, P >0.05. Asterisks indicate significance of telomere length differences in *pop* cells versus WT.

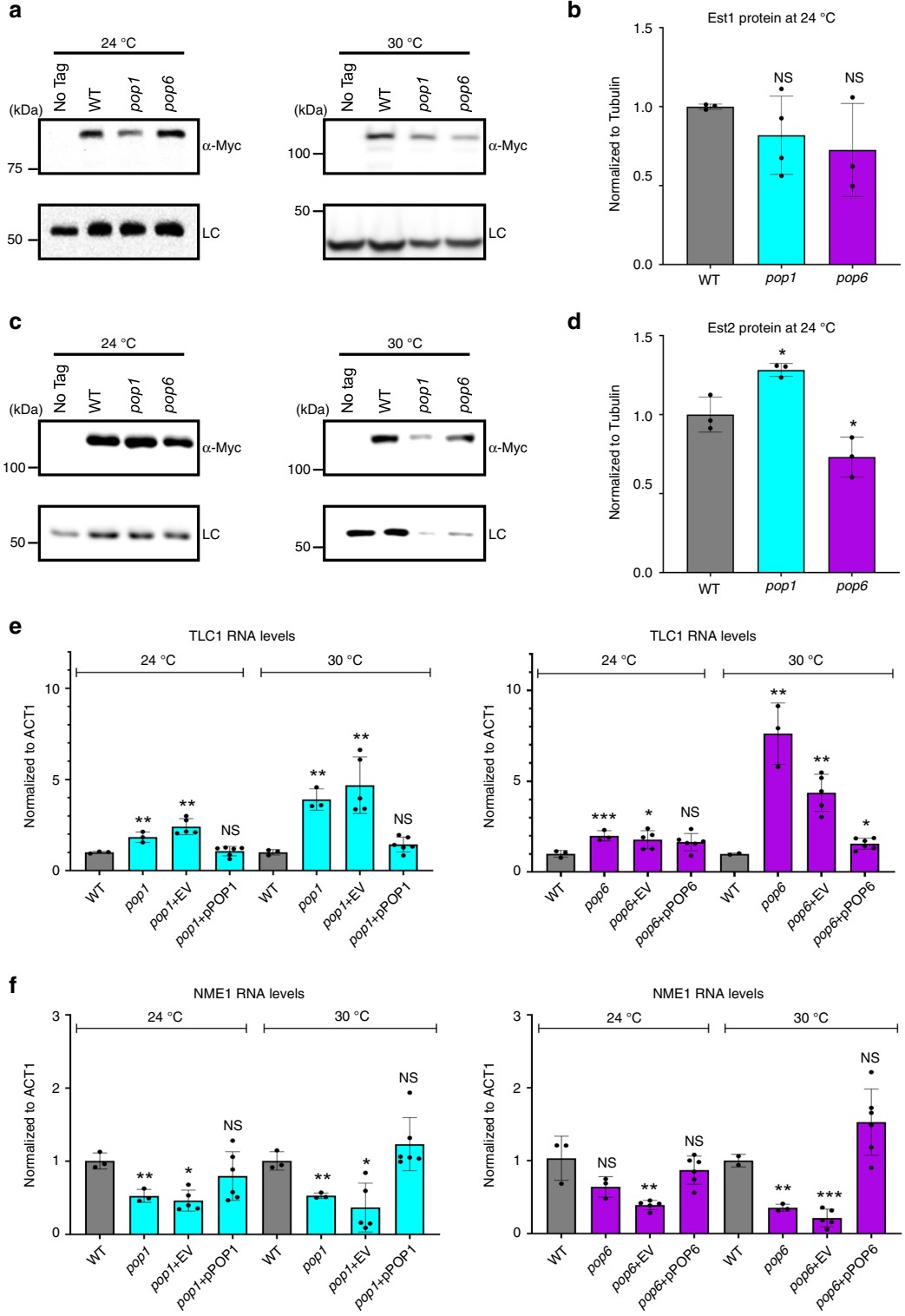

average telomere length in isogenic WT cells was 385 base pairs (bp) at 24 °C and 400 bp at 30 °C. At 24 °C, telomeres in *pop1* and *pop6* cells were short (~0.7× WT; $P < 10^{-3}$, unless otherwise noted, unpaired two-tailed Student's *t*-tests were used to determine statistical significance; see Fig. 3c for more detail). The extent of shortening was even greater at 30 °C (0.6 and 0.5× WT; Fig. 3a, b). The short telomere phenotype was reversed by introducing a plasmid-borne copy of *POP1* or *POP6*. Unlike telomerase deficient cells, the length of *pop* telomeres was stable, and cells did not senesce. Because telomere shortening was

evident at 24 °C when growth rates and protein levels were similar in WT and mutant cells, it was likely due to reduced abundance and/or function of Pop1 and/or Pop6. We conclude that Pop1 and Pop6 are required for WT telomere length.

**Short telomeres in *pop* cells are not due to low Est1 or 2.** To determine if the abundance of key telomerase subunits are Pop protein-dependent, we did western blots on WT and *pop* cells expressing epitope tagged Est1 or Est2 (Fig. 4a–d). At 24 °C, Est1

**Fig. 4 At 24 °C, TLC1 levels are higher in *pop* cells, while Est1 and Est2 levels are similar to WT.** Protein extracts from the indicated strains grown at 24° or 30 °C were analyzed by western blotting using a MYC-antibody to detect **a** Est1-MYC or **c** Est2-MYC. [(Est3 levels were not determined because we do not have an epitope tagged allele of Est3 that has WT telomere length and that can be detected by westerns in whole cell extracts[4]]. Levels of **b** Est1-MYC and **d** Est2-MYC from three biological replicates (black circles) grown at 24 °C were quantified after being normalized to levels of tubulin in the same samples. The normalized value of each protein in WT cells was defined as one. At 30 °C, levels of Est1-MYC and Est2-MYC in *pop* cells were normalized only to WT levels of Est1 or Est2 as tubulin and Pgk1 levels were also lower at 30 °C. The fold difference in Est1-MYC or Est2-MYC in *pop1* and *pop6* cells compared to WT is shown. The loading control (LC) for the Est1-MYC western at 30 °C was Pgk1. For all other westerns, the LC was tubulin. **e** Total RNA was extracted from cells grown at 24 °C or 30 °C for ~50 generations. RNA levels were determined by RT-qPCR for **e** TLC1 or **f** NME1 in WT (gray bars), *pop1* (blue bars) and *pop6* (purple bars). Individual biological replicates are shown (black circles). TLC1 and NME1 levels were normalized to levels of ACT1 mRNA in the same samples using the $2^{-\Delta\Delta Ct}$ method[56]. At both 24 and 30 °C, the increase in TLC1 (and the decrease in NME1) RNA in *pop* cells was suppressed by a WT copy of the mutated *POP* gene. Error bars are one standard deviation from the average value of three or more independent experiments. *P*-values were calculated using unpaired two-tailed Student's *t*-test; *$P \le 0.05$, **$P \le 0.01$, ***$P \le 0.001$, ****$P \le 0.0001$; NS, not significant, $P > 0.05$. Source data are provided as a Source Data file.

levels were not significantly different from WT in both *pop* strains ($P = 0.19$) (Fig. 4a, b). Compared to WT levels, Est2 abundance was modestly increased in *pop1* (1.3× WT, $P = 0.01$) and decreased in *pop6* cells (0.7× WT, $P = 0.05$; Fig. 4c, d). At 30 °C, Est1 and Est2 levels in both *pop* strains were ~0.3× and ~0.6× WT levels, respectively (the reduction in Est2 levels was significant only in *pop1* cells; Fig. 4b, d). The abundance of the loading control was similarly reduced (~0.5× WT; Fig. 4a, c). As there were no significant trends in Est1 or Est2 levels that correlate with short telomeres, we conclude that short telomeres in 24 °C grown *pop* cells are not explained by reduced Est1 or Est2.

**TLC1 levels are high and NME1 levels are low in *pop* cells.** Pop proteins bind directly to their RNA targets: TLC1, NME1 and RPR1[22,27]. NME1 and RPR1 are less abundant in Pop-deficient cells[28]. To determine if TLC1 levels were similarly reduced, we used reverse transcriptase quantitative PCR (RT-qPCR) (Fig. 4e). At 24 °C, TLC1 levels were 2× higher in both *pop* strains ($P < 0.05$). This increase was even higher at 30 °C (4–8× WT; $P < 0.01$; Fig. 4e). In contrast, at 24 °C, NME1 was lower in both *pop* strains (~0.4–0.7× WT; $P < 0.03$; Fig. 4f). NME1 levels were even lower in 30 °C grown *pop* cells (~0.25× WT; $P < 0.001$). Thus, the short telomere phenotypes of *pop1* and *pop6* cells are not due to reduced TLC1. Unexpectedly, Pop proteins have different effects on NME1 and TLC1, even though they bind to regions with similar sequence and structure in the two RNAs[22].

**Short telomeres in *pop* cells are not due to titration of Yku.** The Yku heterodimer not only binds TLC1, it also binds telomeres independently of its association with TLC1[3]. This binding protects telomeres from nucleolytic degradation in G1 phase[29,30]. When TLC1 is expressed at very high levels, it titrates Yku from telomeres, resulting in telomere shortening[16]. These results raised the possibility that the short telomeres in *pop* cells might be explained by removal of Yku from telomeres due to increased TLC1 (Supplementary Fig. 2). If *pop* mutants and Yku depletion affect telomere length by the same mechanism, telomeres should be similarly short in double versus single mutants. However, double mutants (*pop1* or *pop6 yku70Δ* cells) had even shorter telomeres compared to any of the single mutants ($P < 0.001$; Supplementary Fig. 2). Thus, the short telomere phenotypes of *pop* mutants are not due to loss of Yku from telomeres.

**Trimethylated TLC1 levels are similar in *pop* and WT cells.** The 2,2,7 trimethylation (TMG) of the TLC1 5′ methylguanosine (m7G) cap is catalyzed in the nucleolus by the Tgs1 methyltransferase[31]. Acquisition of the TMG is probably required for exit from the nucleolus, because TLC1 accumulates there in *tgs1Δ* cells[13]. To monitor the efficiency of trimethylation, we immuno-

precipitated TLC1 with an anti-TMG antibody (Millipore). The amount of TLC1 precipitated was not significantly different in *pop1* or *pop6* versus WT cells at 24 °C (*pop1* $P = 0.15$; *pop6* $P = 0.5$) or 30 °C (Fig. 5; *pop1* $P = 0.3$; *pop6* $P = 0.6$). We conclude that Pop proteins do not affect acquisition of the TMG cap.

**Sequence of 3′ ends of TLC1 is not altered in *pop* cells.** RNase P/MRP use their endonuclease activities to process RNA substrates. Therefore, we determined if the processing of TLC1 was altered in *pop* mutants. TLC1 exists in two forms, a long poly(A) form (~1300 nts) that accounts for a small fraction of the total TLC1 and a shorter poly(A) minus form (~1157 nt) that also lacks ~140 nucleotides from the 3′ end of the long form. The short form, which accounts for most of the TLC1, purifies with active telomerase[12]. There is no known function for the poly(A) form[32].

There are two models to explain the occurrence of poly(A) TLC1. The first model suggests that mature TLC1 is generated by cleavage of the poly(A) isoform[10]. The second model proposes that the 3′ end of the mature TLC1 is generated by a poly(A)-independent transcription termination pathway[32,33]. According to this model, poly(A) TLC1 is a dead-end product generated when termination fails[1,32]. Given that the RNase P/MRP complexes process other RNAs, the first model provides a potentially unifying hypothesis for the effects of Pop proteins on its RNA targets. The hypothesis that Pop proteins are needed to generate active TLC1 from the longer poly(A) form predicts that the fraction of TLC1 in the long form will be higher in 24 °C grown *pop* cells and even higher at 30 °C.

To determine if Pop proteins affect poly(A) addition, we determined the fraction of TLC1 in the two isoforms in *pop* and WT cells by both northern analysis (data not shown) and 3′ RACE (Rapid amplification of cDNA ends). Contrary to the first hypothesis, the ratio of long to short forms of TLC1 was significantly lower in *pop1* and *pop6* cells than in WT cells at 24 °C (Fig. 6b; ratios of ~0.80, WT; ~0.45, *pop* cells; *pop1* $P = 0.01$; *pop6* $P = 0.03$). The ratios were even lower in *pop* cells at 30 °C.

As a further test, we considered the possibility that impaired RNA processing in *pop* cells might alter the sequence of the 3′ ends of TLC1. We ruled out this possibility by sequencing the 3′ ends of all cDNAs shorter than 500 bps (Fig. 6c, d). The distribution of lengths and sequences of the 3′ends were not significantly different in *pop* versus WT cells at 24 °C. Combined with the TMG data, the 5′ and 3′ ends of TLC1 are processed normally in *pop* mutants.

**The Est1-TLC1 association is unstable in *pop* cells at 24 °C.** Est1 and Est2 bind directly to telomerase RNA (Fig. 1a). By RNA immunoprecipitation (RNA-IP), Est1 and Est2 bind at reduced

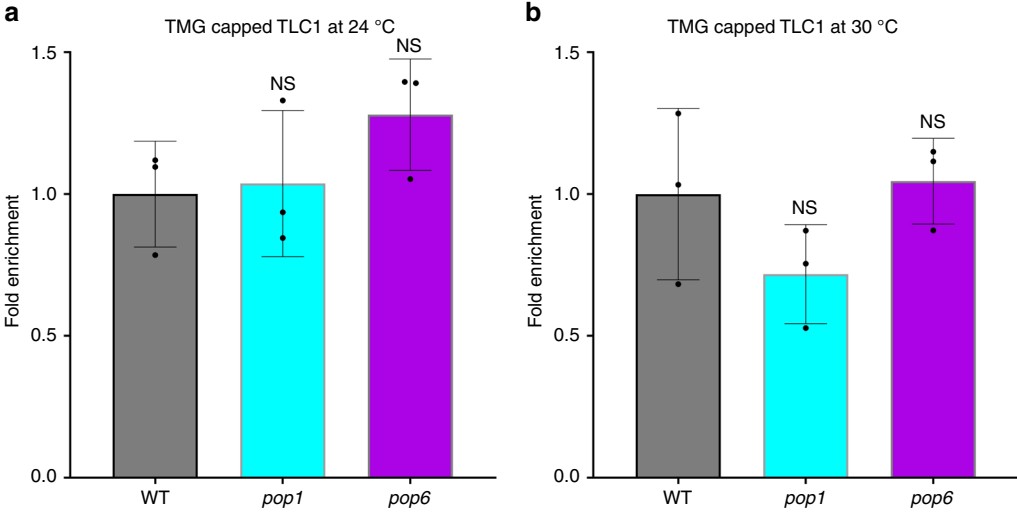

**Fig. 5 Trimethylation of the TLC1 5′ cap is not affected in *pop* cells.** Total RNA was extracted from WT (gray bars), *pop1* (blue bars) and *pop6* (purple bars) cells and TMG-capped TLC1 was immunoprecipitated using an anti-TMG antibody in cells grown at 24 °C (**a**) or 30 °C (**b**). TLC1 in the immunoprecipitate was measured by RT-qPCR. IP/INPUT TLC1 was normalized to IP/INPUT *ACT1* mRNA. Bars are data from three independent biological isolates (black circles). Error bars are one standard deviation from the average value of three or more independent experiments. P-values were calculated using unpaired two-tailed Student's *t*-test; *$P \leq 0.05$, ** $P \leq 0.01$, ***$P \leq 0.001$, ****$P \leq 0.0001$; NS, not significant, $P > 0.05$. Source data are provided as a Source Data file.

levels to a deletion derivative of TLC1[22,25]. This deletion derivative lacks the P3-like CS2a domain to which Pop proteins bind. To determine if the association of WT TLC1 with Est1 or Est2 occurred normally in *pop* cells at 24 °C, we immunoprecipitated Est1 or Est2 from mutant and WT cells. We then did RT-qPCR on the immunoprecipitate (IP) to determine the fraction of TLC1 that was associated with Est1 or Est2. The amount of TLC1 in the Est1 IP was significantly lower than in WT cells (~0.14× WT in both *pop* strains; $P < 5 \times 10^{-4}$; Fig. 7a). This dramatic decrease in Est1-TLC1 binding occurred even though Est1 abundance was similar in *pop* and WT cells (Fig. 4a). Est2 association with TLC1 was reduced significantly only in *pop6* cells (0.6× WT, $P = 0.01$, Fig. 7b).

We also used RNA-IP to determine the level of TLC1 binding to HA-Pop1$_{WT}$ in *pop6* cells and HA-Pop1$_{mut}$ in *pop1* cells (Supplementary Fig. 3). In 24 °C grown cells, TLC1 binding to HA-Pop1$_{mut}$ was only 0.15× of its binding to WT Pop1 ($P = 0.030$). At the same temperature, TLC1 binding to WT Pop1 in *pop6* cells was even lower (0.05× *POP6* cells; $P = 0.007$). Reduced TLC1 binding to Pop1 can be explained by reduced levels of Pop1 in both strains (Fig. 2f, g). We conclude that mutant Pop1 reduces the stable association of Est1 to TLC1 (and in *pop6* cells, Est2). In addition, in combination with the data on Pop1 abundance (Fig. 2f, g), Pop1 stability and its binding to TLC1 are both Pop6-dependent.

**Telomerase binding to telomeres is deficient in *pop* cells.** Results from RNA-IP indicate that the telomerase holoenzyme was unstable in *pop* cells (Fig. 7a, b). This finding predicts that telomerase binding to telomeres would be low in these mutants. To test this hypothesis, chromatin immunoprecipitation (ChIP) was done in 24 °C grown WT, *pop1* and *pop6* cells expressing Est1 or Est2. We determined binding of both proteins to telomeres VI-R and XV-L (Fig. 7c, d). The binding of both Est1 (~0.3× WT, $P < 0.01$) and Est2 (0.7× WT, $P < 0.05$) was significantly reduced at both telomeres in both strains. Impaired stability of the telomerase holoenzyme and its reduced telomere binding likely explain the short telomere phenotypes of *pop1* and *pop6* cells at 24 °C.

**TLC1 folding is WT in *pop* cells except at Pop binding site.** RNA-IP (Fig. 7a, b) revealed that in 24 °C grown *pop* cells, the level of TLC1 association with Est1 was only ~14% of the level in WT cells. In vitro studies suggest that the secondary structures of NME1 and RPR1 are altered in the absence of Pop proteins[24,34]. Therefore, we considered the possibility that the structure of the Est1 and (to a lesser extent) the Est2 binding sites in TLC1 might be Pop protein dependent. This model was appealing as it could provide an explanation for reduced binding of Est proteins to TLC1 in *pop* cells. To test this possibility, we used DMS-MaPseq, a method for high throughput chemical probing of RNA structure in vivo[35,36]. In addition to TLC1, we determined the structure of NME1, which is expected to be Pop protein dependent, in the same RNA samples.

WT and *pop* cells were treated in vivo with (or without) dimethyl sulfate (DMS). DMS methylates unpaired adenosines and cytosines in RNA[37]. The methylation sites were detected using the reverse transcriptase TGIRT III, which introduces mutations, rather than stopping, at methylated sites[38]. The resulting cDNAs were amplified and sequenced by next generation sequencing. Because the ~1157 nt TLC1 was too large to be sequenced as a single amplicon, it was divided into three parts. In total, we sequenced 180 libraries to monitor folding in 60% of the 1157 nt TLC1 and ~75% of the 341 nt NME1. After pre-processing, there were an average of 680,000 reads per A or C. This depth of coverage is sufficient to detect reproducible changes in structure-dependent methylation patterns, even if these changes occur in only a subpopulation of transcripts[39–41].

To test if either TLC1 or NME1 RNAs were differentially reactive to DMS in *pop* versus WT cells, we first calculated the DMS reactivity at each nucleotide. Then, we used dStruct to determine regions with statistically significant differences in reactivities between *pop* and WT cells[41]. dStruct uses a dissimilarity score ($\Delta d$) to assess variation in reactivities within biological replicates for each cell type (*pop* or WT) and between *pop* and WT cells. It then screens for regions where the latter variation exceeds the former and evaluates the significance of such differences. Regions of five nucleotides (or more) that had q-values <0.1 and $\Delta d > 0.01$ were considered statistically significant.

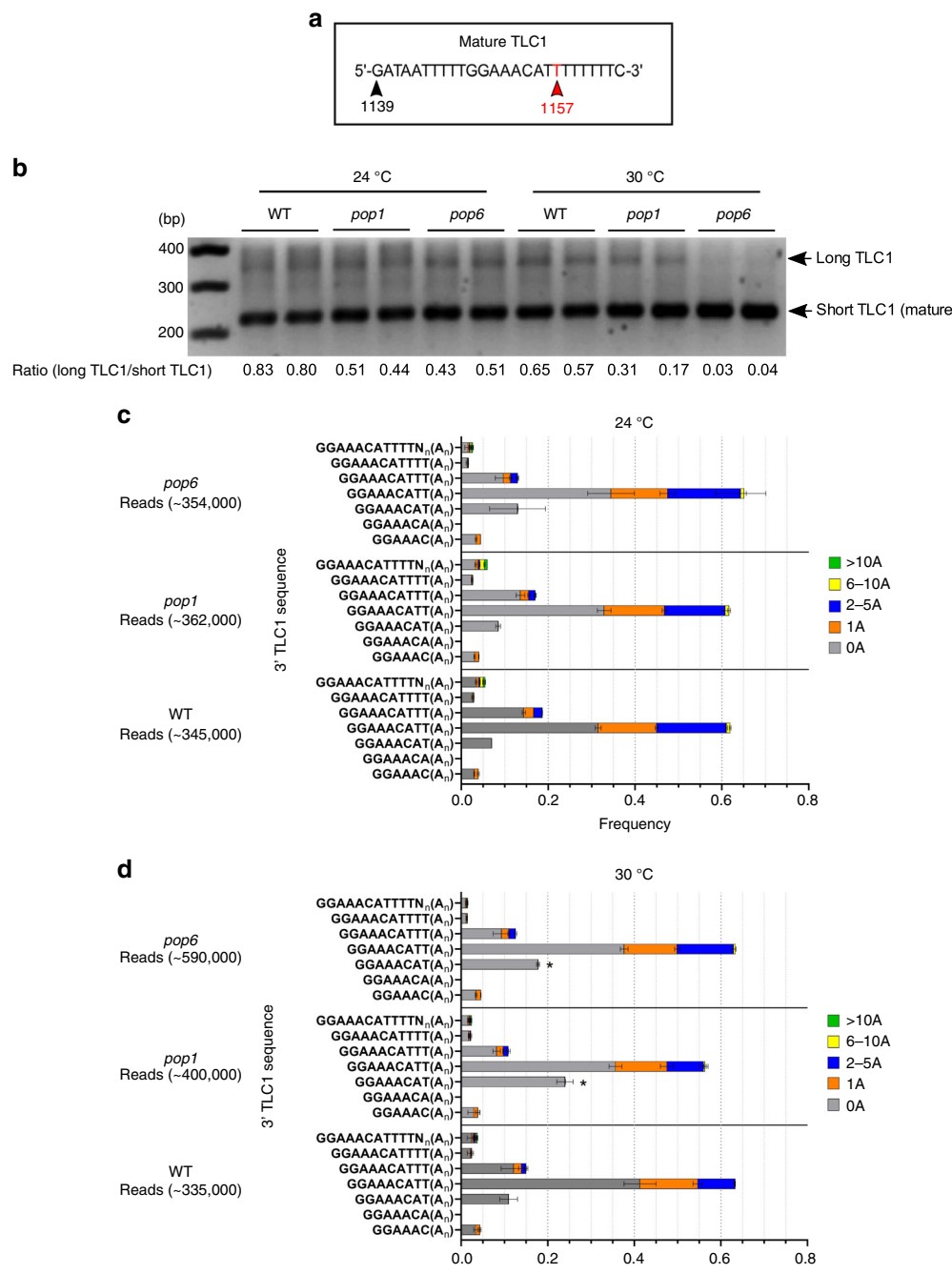

**Fig. 6 Processing of TLC1 3′ ends is not altered in *pop* cells at 24 °C.** The 3′ end sequences of TLC1 extracted from 24 or 30 °C grown cells were analyzed by 3′ RACE. **a** Diagram of the 3′ end of TLC1 from nucleotides 1139 to 1164. The last nucleotide in the 1157 nt mature TLC1 molecule is indicated with a red arrowhead. **b** PCR products amplified from TLC1 cDNA were analyzed on a 1% agarose gel. The expected sizes of mature and precursor forms of TLC1, are, respectively, ~220 and ~350 bp. Compared to WT, the relatively abundance of short to long forms were higher in *pop1* and *pop6* cells at 24 °C and, especially, 30 °C. Ratios of long to mature form are shown below lanes. All amplicons <500 nts shown in panel B were purified and deep sequenced. The relative abundance of each TLC1 species as determined by sequencing was quantified. Sequences that aligned to the 1154 to >1159 nt end of TLC1 with or without a poly(A) tail are shown for cells grown at **c** 24 °C or **d** 30 °C. The size of the poly(A) tails are: no adenosine (gray), 1 adenosine (orange), 2-5 adenosines (dark blue), 6–10 adenosines (yellow) and >10 adenosines (green). Total reads are indicated. Each bar shows the average from biological duplicates. At 30 °C, transcripts ending at nt 1156 [no poly(A)] were significantly more abundant by an unpaired two-tailed Student's *t*-test in *pop1* and *pop6* versus WT (marked with asterisk). Source data are provided as a Source Data file.

As expected, the structure of NME1 was highly Pop sensitive. At 24 °C, there were four regions that had significantly different DMS reactivity in *pop* versus WT cells (Fig. 8a–c; Supplementary Fig. 4). Region one (nucleotides 37–42) localized to the P3 stem loop where Pop6 and Pop7 bind (Fig. 8a, c; Supplementary Fig. 4). The

three other differentially reactive regions were located throughout NME1 (nucleotides 93–121, region two; nucleotides 146–152, region three; nucleotides 247–277, region 4; Fig. 8a, c). Thus, ~26% of the 256 nts analyzed in NME1 met the criteria for differential reactivity in *pop* versus WT cells. At 30 °C, there were two

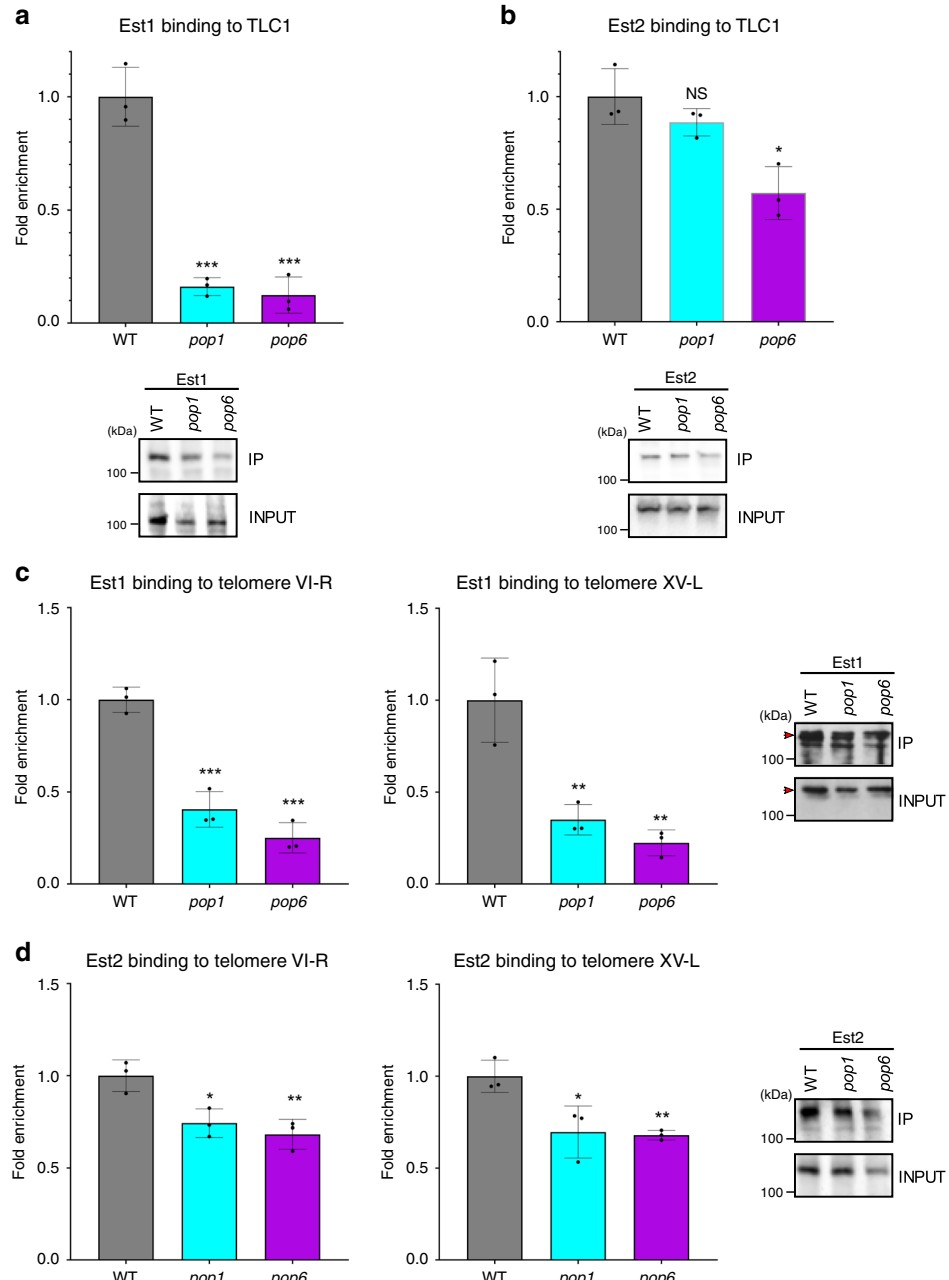

**Fig. 7 Telomerase holoenzyme assembly and telomerase binding to telomeres is deficient in *pop* cells at 24 °C.** RNA immunoprecipitations were done in WT (gray bars), *pop1* (blue bars) and *pop6* (purple bars) cells expressing **a** Est1-MYC or **b** Est2-MYC. Extracts prepared from cells grown at 24 °C for ~50 generations were immuno-precipitated with anti-MYC antibody, and the amount of TLC1 in each precipitate was determined by RT-qPCR. The IP/INPUT levels of TLC1 were normalized to the IP/INPUT levels of *ACT1* mRNA in each sample. The average TLC1/ACT1 ratios in WT cells were defined as one. Fold enrichment and significance in *pop* cells is relative to WT. Chromatin immunoprecipitation assays in WT, *pop1* and *pop6* cells grown at 24 °C for ~50 generations to determine the association of **c** Est1-MYC or **d** Est2-MYC to telomeres VI-R (left) and XV-L (right). The level of telomeric sequence in the IP/INPUT was normalized to the amount of *ARO1* DNA in the IP/INPUT of the same sample. The average ratio of telomeric DNA to *ARO1* in WT cells is defined as one. Fold enrichment and significance in *pop1* and *pop6* cells is relative to WT. Western blots of Est1-MYC or Est2-MYC in both the input and immuno-precipitates (IP) demonstrate that similar amounts of Est1 and Est2 were immuno-precipitated in WT and *pop* cells at 24 °C. Red arrowheads in c) indicate the band for Est1-Myc. Biological replicates in a–d are shown as black circles. Error bars are one standard deviation from the average value of three or more independent experiments. *P*-values were calculated using unpaired two-tailed Student's *t*-test; *$P \leq 0.05$, **$P \leq 0.01$, ***$P \leq 0.001$, ****$P \leq 0.0001$; NS, not significant, $P > 0.05$. Source data are provided as a Source Data file.

additional regions of statistically significant modification in the *pop6* strain (Supplementary Figs. 5–7). Thus, the structure of NME1 RNA is globally altered in *pop1* and *pop6* cells even at 24 °C.

Unexpectedly, TLC1 had a single, short region that was differentially DMS reactive in *pop* versus WT cells at 24 °C (Fig. 8d; Supplementary Fig. 9). The altered region consisted of

only seven nucleotides in the P3-like CS2a domain where Pop6 and Pop7 bind (Fig. 8e). The higher DMS reactivity of these nucleotides can be explained by a reduction in Pop1/6 binding that normally shields them from DMS modification (Fig. 1a). This lower binding is expected as Pop1 was present at ~45% of WT levels in *pop1* and *pop6* cells at 24 °C (Fig. 2f, g). There were

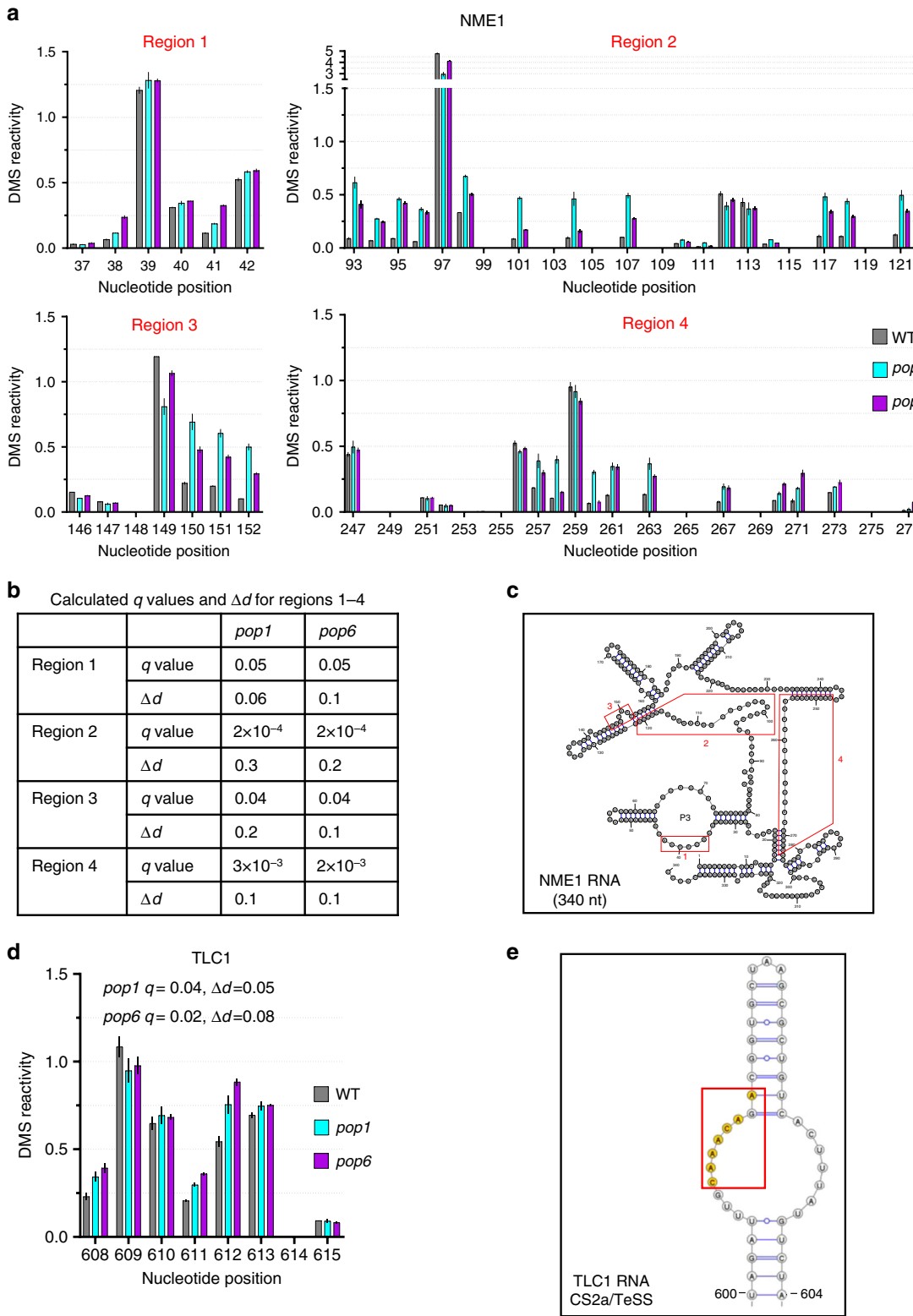

no other significant changes in DMS reactivity in the 710 nucleotides of TLC1 examined, including the binding sites for Est1, Est2, Yku, the $Sm_7$ ring and the template region (Supplementary Figs. 8–10). Thus, fewer than 1% of the analyzed nucleotides in TLC1 met the criteria for being differentially reactive in *pop* versus WT cells compared to 26% in NME1. The WT DMS sensitivity of the Est1 and Est2 binding sites is consistent with the near WT levels of Est1 and Est2 in *pop* cells at 24 °C (Fig. 4a–d).

At 30 °C, there were three overlapping regions with differential reactivity in TLC1 in both *pop* strains, including the binding site for Est1, and two additional regions seen only in *pop1* or *pop6* cells (Supplementary Figs. 11–14). We conclude that Pop proteins affect the global structure of NME1 but not TLC1 at 24 °C. The similarity in DMS-reactivity of the Est1 and Est2 binding sites in *pop* and WT cells at 24 °C suggest that Est1 and Est2 are TLC1-associated at near WT levels in vivo at this temperature. In contrast, the altered DMS reactivity of the Est1 binding site in *pop*

**Fig. 8 The global structure of NME1 but not TLC1 is affected in *pop* cells at 24 °C.** DMS-MaPseq analysis from strains grown at 24 °C for ~50 generations. **a** Four regions of NME1 RNA with statistically significant different DMS reactivities in *pop1* (blue bars) and *pop6* (purple bars) compared to WT (gray bars) cells are shown. DMS reactivities from three independent biological replicates were averaged. **b** q Values and Δd values of regions 1–4 of NME1 RNA in *pop1* and *pop6* compared to WT cells. Regions of five nucleotides (or more) that had q-values < 0.1 and Δd > 0.01 were considered statistically significant. **c** The four regions shown in (A) are mapped onto the NME1 RNA structure[23]; region numbers are the same as in **a**. The significantly altered regions are denoted by red boxes. **d** Average normalized DMS reactivity from nucleotides 608–615 in TLC1. These nucleotides are at the site of Pop6/7 binding. Symbols are the same as in panel A. q values and Δd values of the indicated region in *pop1* and *pop6* compared to WT. **e** Enlarged view of the CS2a/TeSS domain of TLC1 indicated in **d**. Nucleotides in orange indicate the only nts in TLC1 that met the criteria for being differentially reactive. See supplementary Figs. 4–7 for DMS reactivity in regions where there was no significant difference between *pop* mutants and WT. Source data are provided as a Source Data file.

cells at 30 °C, suggests that binding of Est1 was reduced at the higher temperature. This conclusion is consistent with the reduced abundance of Est1 at 30 °C (Fig. 4a, c).

At first glance, the 24 °C data from RNA-IP (Fig. 7a,b) and DMS-MaPseq (Fig. 8; Supplementary Figs. 8–10) appear contradictory. By RNA-IP, only ~14% of TLC1 RNA was Est1 associated. However, by DMS-MaPseq, the Est1 binding site had a WT structure suggesting similar levels of Est1 binding to TLC1 in *pop* and WT cells at 24 °C. As detailed in the discussion, the two methods monitor different aspects of the Est1-TLC1 association. DMS-MaPseq monitors Est1-TLC1 binding in vivo. RNA-IP detects Est1-TLC1 associations that are sufficiently stable to persist for the time needed to process the cellular lysate.

**TLC1 re-localization to the nucleus is impaired in *pop* cells.** Although an unstable holoenzyme that has reduced telomere binding can explain the short telomere phenotype of *pop* cells, it does not explain the increased abundance of TLC1. High levels of TLC1 are not due to faulty RNA processing (Figs. 5, 6) nor to its global mis-folding (Fig. 8; Supplementary Figs. 8–10). However, if an unstable holoenzyme is less likely to be imported back to the nucleus, TLC1 might accumulate in the cytoplasm where it would not be accessible to the nuclear exonucleases that degrade it[42]. To test this possibility, we used RNA FISH to determine if Pop deficiency altered the sub-cellular distribution of TLC1.

To detect TLC1, we used a Cy3-labeled (green) 52-nts long DNA probe complementary to nts 389 to 414 of TLC1. The probe contains a 26 nts sequence that binds TLC1 including a 10-nts toehold and a 16-nts stem that forms a hairpin-like structure with a 10-nts loop (Fig. 9a)[43]. When the 10 nts portion binds TLC1, the stem region unzips only for a fully matching RNA molecule allowing stable hybridization (Fig. 9a). A probe containing a hairpin has a higher target specificity than an unmasked probe with the same cognate sequence[1,43,44] (Supplementary Figs. 15 and 16). The nucleolus was detected with a similarly structured Cy5-labeled (red) FISH probe that targeted the transcribed spacer ITS1 in the 35s rRNA precursor RNA[45]. The specificity of the TLC1 probe was demonstrated by its very low signal in *tlc1Δ* cells (Supplementary Fig. 17). FISH spots arose predominately from single RNA molecules as evident from single-step photobleaching (Supplementary Fig. 18). The nucleus was stained with DAPI, and cell size was determined by DIC microscopy. (See Fig. 9a and Supplementary Fig. 19 for representative images.) TLC1 spots that colocalized with Cy5 and DAPI were categorized as nucleolar and nuclear, respectively. Spots that colocalized with neither but were present within the boundaries of a cell were categorized as cytoplasmic. Our values for TLC1 distribution in nuclear versus cytoplasmic fractions were similar to those reported by others for WT cells when our values for TLC1 in the nucleolar and nucleoplasmic fractions were combined[46].

The average number of Cy3-labeled FISH spots per cell at 24 °C was ~9 (WT), 23 (*pop1*; 2.7× WT), and 16 (*pop6*; 1.9× WT) (Fig. 9b, d). For each strain, the number of TLC1 spots was higher

at 30 °C than at 24 °C (10.5, WT; 31 or 3× WT, *pop1*; 48 or 4.6× WT, *pop6*). The fold increase in TLC1 RNA in *pop* compared to WT cells was significant at both temperatures (all *P*-values <0.005) and remarkably consistent with fold increases in TLC1 as determined by RT-qPCR (Fig. 4e). The number of detected TLC1 FISH spots per WT cell (~11) is smaller than the number of TLC1 molecules per cell (29) as determined by a biochemical assay[8]. This lower number agrees with previous studies that also showed substoichiometric detection efficiency with this FISH protocol[47].

We determined the number of TLC1 molecules in the nucleoplasm, the nucleolus and the cytoplasm in WT and *pop* cells at 24° and 30 °C (Fig. 9c, d). Compared to WT, the fraction of TLC1 molecules in the nucleolus in *pop* cells at both 24 °C and 30 °C was not significantly different from WT (Fig. 9d). However, in both *pop* strains, there was a significantly larger fraction of TLC1 molecules in the cytoplasm compared to WT cells. This difference was significant at both 24 °C and 30 °C but was exacerbated at 30 °C (fraction of cytoplasmic TLC1: 0.32, WT vs ~0.40, *pop* cells at 24 °C; 0.28, WT vs ~0.52, *pop* cells at 30 °C). We conclude that the transit of TLC1 from the cytoplasm back to the nucleus is impaired in *pop1* and *pop6* cells.

## Discussion
Pop proteins have been known for decades owing to their essential functions as protein subunits of two RNA-multi protein complexes, RNAse P and MRP. Here, we examine the significance of the unexpected finding that Pop1, Pop6, and Pop7 are telomerase associated[5,22]. RNase P, the better studied of the two complexes, is mostly known for its ubiquitous role in tRNA processing. However, non-tRNA targets have been identified in bacteria, yeast, and humans[23]. For example, RNase P cleavage generates the mature 3′ end of the human MALAT1 non-coding RNA that is mis-regulated in many cancers[48]. The role of Pop proteins on TLC1 is fundamentally different from the role of RNase P at other atypical substrates, as it involves only three of the nine protein subunits of RNase P[5]. Moreover, Pop proteins did not affect the processing of TLC1 as formation of the 5′ and 3′ ends of TLC1 were unchanged in *pop* cells, even at 30 °C (Figs. 5 and 6). These results raise the possibility that Pop proteins may act non-catalytically on other RNA-protein complexes.

To study how Pop proteins affect TLC1, we reduced the abundance and activity of Pop1 or Pop6 using temperature sensitive alleles (Fig. 2a). At 24 °C, *pop1* and *pop6* cells grew about as well as WT (Fig. 2c), and the abundance of many proteins (Fig. 2e), including telomerase subunits Est1 and Est2 (Fig. 4a–e), were similar to WT levels. In contrast, at 24 °C, mutant Pop1 was present at ~45% of WT levels. Studying *pop* mutants at permissive temperature almost surely underestimates their impact on telomeres and telomerase. However, this strategy makes it likely that telomerase defects are due to direct effects of altered Pop proteins rather than to a general decline in protein synthesis.`Even though our approach was conservative, we detected multiple, statistically

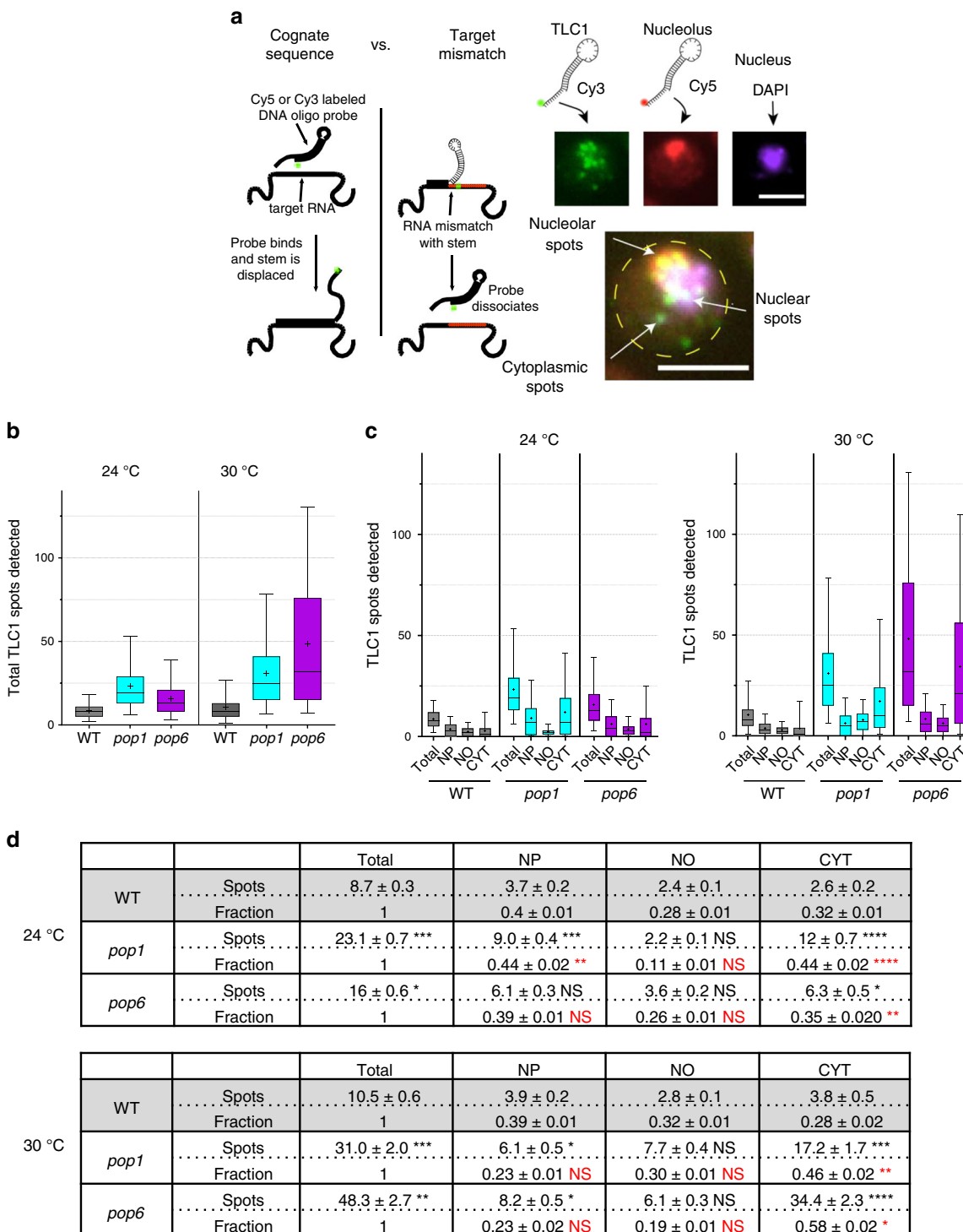

**Fig. 9 Re-entry of TLC1 RNA into the nucleus is impaired in *pop* cells. a** Schematic of the FISH method. FISH probes have a hairpin structure. Specificity is enhanced by strand displacement of the stem by the cognate RNA, but any mismatch in the target inhibits initial binding or strand displacement. TLC1 (green) is targeted by a short 52-nt DNA Cy3-labeled probe; ITS1 (red), nucleolar RNA, is targeted with a Cy5-labeled 52-nt oligo probe. The nucleus (purple) is visualized with DAPI. A composite image of a *pop6* cell is shown from the overlap of 3 channels. TLC1 spots are assigned a localization based on the overlap with the intensity in the other two channels. Scale bar: 5 µm. **b** Total number of FISH spots per cell in cells grown at 24 °C and 30 °C. The number of cells analyzed per strain ranged from 328 to 538. **c** Spot distributions for subcellular compartments. Nucleoplasm (NP), Nucleolus (NO), Cytoplasm (CYT). Whiskers on distributions represent the 5 and 95% percentiles, the box extends from 25 to 75%, the middle of the box represents the median and the + sign represents the mean. **d** Spots and fractions indicate the mean and standard error of FISH TLC1 spots per cell. Fractions were calculated by normalizing the TLC1 spots in each subcellular compartment by the total number of TLC1 spots per cell. P values were calculated using the Kolmogorov-Smirnoff two sample test. Black asterisks indicate significance of difference in spot numbers in *pop1* or *6* versus WT cells; red asterisks indicate significance of difference in fraction of spots in *pop1* or *6* versus WT cells. Because the fraction of TLC1 in the NO was low, a two-fold fractional change in nucleolar TLC1 in *pop1* versus WT at 24 °C was not significant. Source data are provided as a Source Data file.

significant effects on the abundance and structure of TLC1 and NME1.

Mutation of either Pop1 or Pop6 led to dramatic effects on telomeres and telomerase at 24 °C. These effects were exacerbated at 30 °C. Compared to WT, telomere lengths at 24 °C were reduced by ~27–40% (30–50% reduction at 30 °C) (Fig. 3). Short telomeres were not due to reduced levels of three key telomerase subunits. At 24 °C, Est1 and Est2 were present at close to WT levels, while levels of TLC1 were twice as high in *pop* compared to WT cells (4–6× WT at 30 °C; Fig. 4e). However, like tubulin and Pgk1 (Fig. 2e, f), Est1/2 levels were reduced at 30 °C (0.4× WT) (Fig. 4a, c).

Reduced Pop proteins had very different effects on TLC1 versus NME1 RNA. While levels of TLC1 were elevated, NME1 was reduced to ~50% of its WT levels in *pop* cells at 24 °C (25% of WT at 30 °C; Fig. 4f). The reduction in NME1 is likely due to the global disruption of its structure, as by DMS-MaPseq analysis, 26% of the examined C and A's in NME1 were differentially DMS-sensitive at 24 °C (Fig. 8). These structural changes probably target NME1 for degradation. In contrast, only seven nts in TLC1, fewer than 1% of those examined, were differentially reactive to DMS at 24 °C in *pop* cells (Fig. 8d; Supplementary Figs. 8–10). These seven nucleotides are the site of Pop binding[22]. Their higher reactivity is easily explained by their being less protected from DMS in *pop* cells owing to the reduction and/or altered activity of Pop proteins (Fig. 2f, g). In contrast, the Est1 binding site, which is very close to the Pop binding site, was not altered by Pop deficiency at 24 °C (Supplementary Fig. 9). Likewise, the Est2, Yku, and Sm$_7$ binding sites had WT DMS sensitivity (Supplementary Figs. 8–10). Although it is possible that some changes in DMS accessibility were below detection, the large number of reads, reproducibility between replicates and statistical methods allow detection of differences that occur in only a subset of molecules[39–41]. Moreover, using the same DMS-MaPseq method, the Est1 binding site in TLC1 was differentially DMS-sensitive at 30 °C (Supplementary Figs. 11–14). At this temperature, Est1 levels were reduced (0.4× WT, Fig. 4b, d). The altered DMS reactivity of the Est1 binding site at 30 °C is probably explained by this site being less protected from DMS owing to lower binding by Est1 (Supplementary Figs. 12 and 14). Taken together, the DMS-MaPseq data indicate that Est1 and Est2 are TLC1-associated to similar extents in WT and *pop* cells at 24 °C.

We also studied Est1 and Est2 binding to TLC1 using RNA-IP (Fig. 7a, b). By this assay, the fraction of TLC1 molecules that were Est1 associated in *pop* cells at 24 °C was only ~14% of WT (Fig. 7a). By the RNA-IP assay, Est2-TLC1 binding was also reduced, but the reduction was significant only in *pop6* cells (~60% of WT; Fig. 7b). Telomerase binding to telomeres was also significantly impaired (~70% of WT at 24 °C; Fig. 7c, d). The more dramatic effects of Pop proteins on Est1 (compared to Est2) could be related to its cell cycle regulated abundance and telomere binding[2,3]. In contrast, Est2 levels are fairly constant, and its association with TLC1 and telomeres occurs throughout much of the cell cycle. In addition, in the two-dimensional TLC1 structure (Fig. 1a), the Pop binding site is much closer to the Est1 than to the Est2 binding site. This greater proximity probably also contributes to the greater Pop-sensitivity of Est1.

The results from DMS-MaPseq (Fig. 8, Supplementary Figs. 8–10), which suggest WT levels of Est1 and Est2 binding to TLC1 in *pop* cells at 24 °C seem at odds with the results from RNA-IP (Fig. 7a, b). However, the two assays are very different. DMS modifies RNAs in living cells, whereas, the antibody in RNA-IP is added to a lysate. There is no cross-linking in RNA-IP to preserve protein-RNA interactions. If a protein dissociates from TLC1 during isolation, it is unlikely to rebind TLC1 in the lysate. Therefore, DMS accessibility reveals Est1 (and Est2)

occupancy on TLC1 at the time the DMS is added, while RNA-IP detects only stable Est1/2-TLC1 interactions. The differences in the data in the two experiments are best explained by a model in which Pop proteins stabilize the Est1-TLC1 (and to a lesser extent, the Est2-TLC1) interaction, rather than being required for Est1/2-TLC1 binding. A previous study also used RNA-IP to investigate the association of Est1 and Est2 to a deletion derivative of TLC1[22]. However, this work did not distinguish a role for Pop proteins in Est1/2-TLC1 binding from a role in stabilization of Est1/2-TLC1 interactions. We conclude that Pop deficiency results in reduced stability of the holoenzyme due to unstable association of Est1 (and hence Est3 and to a much lesser extent Est2) with TLC1. As a result, the holoenzyme is less likely to be telomere-associated and telomeres are short. This interpretation also provides a plausible explanation for the stimulating effects of Pop1 on in vitro telomerase assays[22]. Telomerase activity is similar in extracts from WT, *est1Δ* and *est3Δ* cells[49]. This result is hard to understand given that Est1 is essential to activate telomerase in vivo[18]. The stimulating effects of Pop1 on an in vitro telomerase assay can be explained by its stabilizing the association of Est1 and hence Est3 (and to a lesser extent Est2) with telomerase. This stimulation should render the normally Est1-independent in vitro reaction Est1-sensitive. This hypothesis could be tested by adding Pop1 to a telomerase extract prepared from *est1Δ* cells.

Although a role for Pop proteins in stabilizing Est1-TLC1 interactions can explain the short telomere phenotype of *pop* cells, the increase in TLC1 is hard to understand with this model. Because mature TLC1 associates with Est proteins in the cytoplasm[13,15], we considered that transit of TLC1 back to the nucleus might be impaired in *pop* cells. We used RNA FISH to monitor TLC1 localization using atypically structured probes that increase the specificity of the FISH signal[47] (Fig. 9). The fraction of TLC1 in the nucleolus was similar in *pop* versus WT cells at 24° and 30 °C. Thus, consistent with no change in TMG capping (Fig. 5), movement in and out of the nucleolus, did not appear to be affected by Pop deficiency. Although at 24 °C, TLC1 was present in the nucleoplasm in both *pop* strains, it was significantly over-represented in the cytoplasm in *pop1* and *pop6* versus WT cells (Fig. 9). This high cytoplasmic localization was exacerbated at 30 °C. Elevated levels of mature TLC1 in *pop* cells (Fig. 6) support the conclusion from RNA FISH that it accumulates in the cytoplasm, as longer residence in the cytoplasm would protect it from nucleases allowing its copy number to increase. Thus, this work identifies an additional requirement for efficient TLC1 re-entry into the nucleus. That is, when Pop proteins are limiting, TLC1 accumulates in the cytoplasm despite the presence of WT levels of Est proteins and nuclear export factors.

We propose the following model (Fig. 10). Most steps in TLC1 biogenesis, such as RNA processing, occur normally in Pop-deficient cells. However, Pop proteins are required to stabilize the association of Est1 (and hence) Est3 with TLC1. The stable Est2-TLC1 association is also promoted by Pop proteins but to a lesser extent. We propose that the import factors that escort telomerase back to the nucleus in late S/G2 phase recognize some aspect(s) of the holoenzyme. In contrast, the Yku-mediated G1 phase import pathway allows nuclear import of Est2-TLC1-Yku because this pathway does not depend on formation of a stable holoenzyme. By this model, the increase in TLC1 abundance is a secondary consequence of TLC1 mis-localization as the RNase activities that are thought to degrade TLC1 are nuclear-localized (Trf4, a component of TRAMP, and Rrp6, a component of the exosome)[42]. Thus, sequestering TLC1 in the cytoplasm protects it from degradation, resulting in increased copy number.

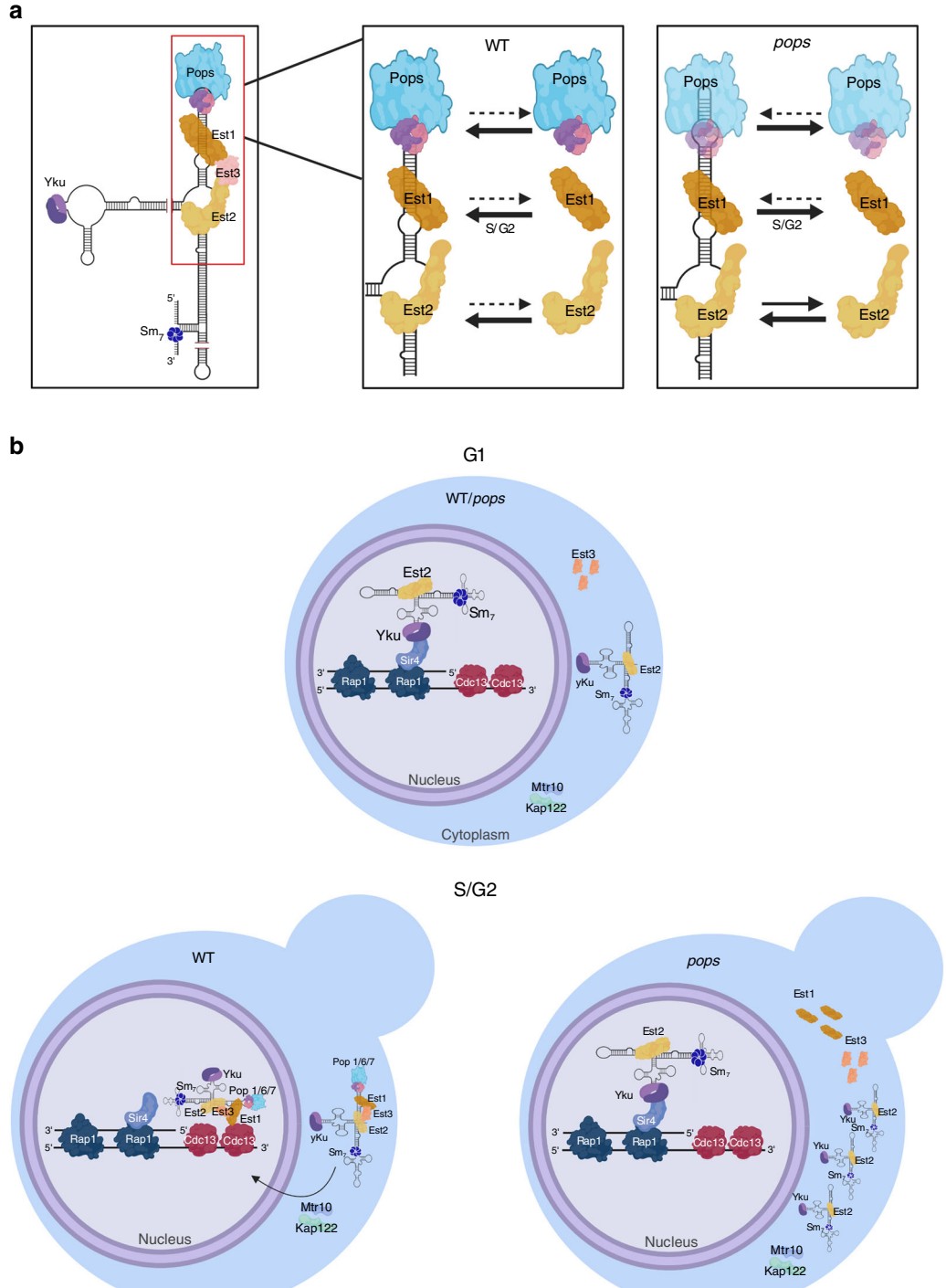

**Fig. 10 Working model for how Pop proteins affect the maturation and localization of telomerase. a** Our data indicate that the Pop proteins are important to stabilize the binding of Est1 (and hence Est3) (and to a lesser extent Est2) to TLC1 RNA. As the cell progresses into late S phase, reduced levels of Pop1 or Pop6 result in lower amounts of stable holoenzyme. We propose that the import factors that promote the re-entry of TLC1 and its associated proteins to the nucleus recognize only the holoenzyme. As a result, TLC1 accumulates in the cytoplasm in *pop* cells. **b** In WT cells, during G1 phase, TLC1 RNA binds to telomeric chromatin via a TLC1-Yku/Sir4/Rap1 interaction[3,18]. The G1 phase binding of telomerase to telomeres is less stable than its association in late S/G2 phase[20]. Chromatin bound G1 telomerase is unable to lengthen telomeres as it lacks two essential subunits, Est1 and Est3[2,4], and is not engaged at the chromosome end[19]. The abundance of core telomerase subunits is fairly constant throughout the cell cycle except for Est1, whose abundance is low in G1 and peaks in late S/G2 phase[2,4,6] owing to ubiquitin mediated proteolysis[5,6]. Est1 is required to recruit Est3 and form the telomerase holoenzyme[4,6]. Therefore, the telomerase holoenzyme, which assembles in the cytoplasm, is present only in late S/G2 phase. The holoenzyme is transported to the nucleus with the help of import factors such as Mtr10 and Kap122[13,14]. In late S/G2 phase, telomerase engages the single-strand G-tails via a Cdc13-Est1 interaction. Either the G1 or the late S/G2 recruitment pathway is sufficient to maintain telomeric DNA, but the Cdc13/Est1 interaction is essential for telomerase activation[18]. In both WT and *pop* cells, the TLC1-Ku/Sir4/Rap1 interactions brings Est2 to telomeres during G1 phase. In late S/G2 phase, reduced levels and/or activity of Pop proteins, lead to instability of the telomerase holoenzyme. The unstable holoenzyme is less likely to be bound by import factors which results in its cytoplasmic accumulation. Images were made in ©BioRender (biorender.com).

Although the RNase P/MRP components and their roles in RNA processing have been studied for decades, they have only recently been linked to telomerase[5,22]. As shown here, Pop proteins affect NME1 and TLC1 by very different mechanisms despite the functional interchangeability of the regions to which they bind in the two RNAs[22]. Previous roles for Pop proteins, even on non-canonical substrates, involve RNA processing, yet Pop proteins affect yeast telomerase by a non-catalytic mechanism. The same MS approach that discovered the telomerase association of *S. cerevisiae* Pop1, 6 and 7[5] found that *Schizosaccharomyces pombe* Pop1 also co-purifies with epitope-tagged telomerase[50]. Thus, Pop proteins may have conserved functions in telomerase regulation.

## Methods

**Yeast strains and methods**. *S. cerevisiae* strains, DNA primers for PCR, and bacterial plasmids are listed in, respectively, Supplementary table 1 (yeast), 2 (plasmids), and 3 (primers). All yeast strains were made in the S288C background. Cells were grown in defined complete (YC) media or in YC minus uracil at 24 °C, 30 °C, or 37 °C. Complementation experiments were done using the *URA3* centromere plasmid pRS416 into which was inserted the entire *POP1* or *POP6* gene. For this cloning, genes were amplified from yDG199 genomic DNA using primers DG192 and DG193 for *POP1* and DG194 and DG195 for *POP6*. Restriction sites for XhoI and HindIII were designed at the 5′ ends of these primers. PCR amplification was performed with iProof High-Fidelity Polymerase (BIO-RAD) according to the manufacturer's directions. The sequence of the genes was confirmed by sequencing. Transformations were done using the lithium acetate/PEG method[51]. Transformants were selected on YC minus uracil media and grown for ~100 generations at 24 °C or 30 °C before testing phenotypes. Thirteen MYC epitopes were introduced at the carboxyl termini of Est1 and Est2 proteins using plasmid pFA6a-13MYC-HIS3MX6[52]. To improve the stability of the tagged proteins, a GLY$_8$ linker was introduced at the 5′ end of MYC-HIS3MX6[53]. The following primers were used for tagging: DG298 and DG299 for Est1 and DG300 and DG301 for Est2. Wild-type Pop1 and the mutant form of Pop1 were epitope tagged by adding 3HA at the N-terminus of each protein. WT, *pop1* and *pop6* strains were transformed with 1 μg of a gene block (IDT DNA) that contained homology to the upstream region of *POP1*, the HIS3MX6 gene, 300 bp sequence upstream of the *POP1* start codon, the 3HA tag and ~400 bp downstream of the *POP1* start codon. Primers used to integrate the 13MYC tag are listed in Supplementary Table 3. To measure growth rates, WT (yDG199), *pop1* (yDG190) or *pop6* (yDG187) cells were grown at the desired temperatures in YC media to an $OD_{600nm} = 0.5$, then diluted back to $OD_{600nm} = 0.01$ in prewarmed YC media. The $OD_{600nm}$ was measured every 2 h for 24 h.

**Southern blots**. For telomere length analysis, WT, *pop* and *ku70Δ* strains were grown on YC or YC minus uracil plates at 24 °C or 30 °C for ~100 generations. Three colonies were chosen from each strain and grown in liquid media at 24 °C or 30 °C. Genomic DNA was extracted from 1.5 mls of saturated cultures $OD_{600nm} = 2.5$ using the MasterPure DNA purification Kit (Lucigen MPY80200) as directed by the manufacturer. Southern blots of telomeric DNA were done by extracting 12.5 mls of genomic DNA followed by overnight digestion with XhoI (New England laboratories). XhoI digested DNA and 5 ng of 1 kb Plus DNA ladder (Invitrogen 10787-018) were run on a 1% agarose gel in TBE buffer (45 mM Tris-borate/1 mM EDTA), transferred to an Amersham hybond XL membrane (GE Healthcare 106 00003), UV crosslinked and hybridized to a telomeric probe isolated from EcoRI digested pCT300 plasmid[54]. The probe was radiolabeled using the Amersham Rediprime Labelling system (GE Healthcare RPN1633). Blots were visualized using a Typhoon FLA 9500 (GE Healthcare).

**Western blots**. Protein extraction was done using the whole cell extract method[55] with some modifications. Briefly, proteins were extracted from 10 mls of culture grown to $OD_{600nm} = 0.5$ at 24 °C or 30 °C for ~35 generations and cells were washed in CE lysis buffer (50 mM HEPES pH 7.5, 140 mM NaCl, 1 mM EDTA pH 8.0, 10% Glycerol, 1 mM DTT and complete, Mini, EDTA-free Protease Inhibitor Cocktail (Roche) and resuspended in 200 μls of CE lysis buffer. Yeast cells were disrupted with glass beads and extracts were resuspended in 50 μls of 2x SDS-PAGE sample buffer (4% SDS, 20% glycerol, 120 mM Tris-HCl pH 6.8, 0.02% bromophenol blue and 5% 2-mercaptoethanol). The samples were then boiled at 95 °C for 5 min, pelleted, and 10 μls of the supernatant were loaded onto an 8% acrylamide SDS-PAGE gel. The proteins were transferred to a nitrocellulose membrane (GE Healthcare 106-00063). MYC tagged proteins were detected with a monoclonal c-MYC antibody (1:400 dilution; Takara 631206) and HA tagged proteins were detected with a monoclonal HA antibody (1:1000 dilution; Santa cruz sc-7392). The controls were detected using a monoclonal anti-α-tubulin antibody (1:1000 dilution, Sigma T6074) and anti-Pgk1 (1:1000 dilution; Abcam ab113687). Protein detection was done using ECL Prime detection reagents (GE

healthcare) according to the manufacturer's directions. The signal was visualized using a FluorChem HD2 Alpha EaseFc system and protein quantification was done using Image J. Western blots of all replicates are shown in the source data file.

**RNA extraction and reverse-transcriptase qPCR**. Total RNA was extracted from 5 mls of cells grown at $OD_{600nm} = 0.5$–0.7 using 400 μls of TES buffer (10 mM Tris, 10 mM EDTA, 0.5% SDS), hot acidic phenol (pH 4.3) saturated with 0.1 M citrate buffer (Sigma-Aldrich P4682), ethanol precipitated and eluted in RNase free water. Five μg of RNA was subsequently DNase treated with Turbo DNA-free (Invitrogen AM107) according to the manufacturer's instructions and eluted with 50 μl of elution buffer. cDNA was synthesized from 100 ng of DNase treated RNA using the iTaq Universal SYBR green one-step kit (BIO-RAD 172-5151) and analyzed using a BIO-RAD CFX96 real-time system. The primers used to measure TLC1 (DG293 and DG294), ACT1 (DG52 and DG53) and NME1 (DG330 and DG331) RNAs are listed in Supplementary Table 3. Changes in RNA expression were analyzed from biological triplicates using the $2^{-\Delta\Delta CT}$ method[56].

**RNA immunoprecipitation**. Previously described methods with some modifications[3] were used to measure the binding of Pop1, Est1 and Est2 to TLC1. Briefly, cells expressing HA-Pop1, Est1-MYC or Est2-MYC were collected from 250 mls of culture grown to $OD_{600nm} = 0.5$ at 24 °C. Cell lysis was accomplished with glass-beads in TMG100 buffer (10 mM Tris-Cl pH8, 1 mM MgCl₂, 10% glycerol, 100 mM NaCl, 0.1 mM EDTA, 0.1 mM DTT). A complete mini EDTA-free protease inhibitor tablet, 20 μls of RNasin Plus RNase Inhibitor (Promega N2618), and 20 μls of SUPERase In RNase Inhibitor (Invitrogen AM2696) were added to 10 mls of TMG100 buffer. For immunoprecipitations, 500 μls of TMG100 plus inhibitors, 0.5% Tween 20 and 10 μls of monoclonal c-MYC antibody (Takara 631206) or anti-HA (Santa Cruz sc-7392) were added to 1 mg of total protein and incubated overnight at 4 °C. Dynabeads protein G (Invitrogen) were equilibrated with TMG100 and 0.5% Tween-20, added to the samples and incubated for 4 h at 4 °C. After three washes with TMG100 and 0.5% Tween-20 and one wash with TMG100, the beads were resuspended in TMG100. The IP and INPUT samples were treated with proteinase K (Thermo Fisher 2546), and RNA was extracted with hot acidic phenol method and ethanol precipitated as described above. RNA samples were DNased with Turbo DNA-free (Invitrogen AM107) and eluted with 50 μls of TE buffer.

**Chromatin immunoprecipitation**. Chromatin immunoprecipitation (ChIP) was performed as previously described[57]. Biological triplicates for each strain were grown in YC media at 24 °C, and 50 mls of culture ($OD_{600nm} = 0.5$) were crosslinked with 37% formaldehyde for 5 min followed by quenching with 2.5 M glycine for 5 min. Crosslinked samples were lysed with glass-beads using a Fast-prep system (Millipore). Crosslinked DNA was sheared with a Branson 450 sonicator 2×9 pulses and 50% output. The sheared samples were then incubated with 10 μls of monoclonal c-MYC antibody (Takara 631206) for one hour at 4 °C. Immunoprecipitation was done using 80 μls of Dynabeads protein G (Invitrogen) and incubated for 2 h at 4 °C. After the washes, the samples were reversed-crosslinked overnight in 1× TE and 1% SDS at 65 °C. The input and immunoprecipitated (IP) DNA were analyzed by quantitative PCR using iQ SYBR Green Supermix (BIO-RAD) with primer pairs DG358 and DG359 for telomere XV-L and primer pairs DG360 and DG361 for telomere VI-R (Supplementary table 3). The following PCR amplification conditions were used: an initial denaturation of 95 °C for 3 min, and 40 cycles at 95 °C for 10 s, annealing temperature of 46–55 °C for 30 s, and extension at 72 °C for 15 s.

The IP/Input data was normalized to an internal control ARO1 using primers DG362 and DG363. The protein levels of IP and Input samples were analyzed by Western blotting using monoclonal c-MYC antibodies as described above.

**Targeted DMS-MaPseq and library preparation**. To analyze the structures of TLC1, NME1, and ASH1 RNAs in WT and *pop* cells, log phase cultures of WT and *pop* mutants were treated with dimethyl sulfate (DMS; D186309-5ML, Sigma Aldrich), and the RNA was analyzed by targeted RNA sequencing as previously described[36] with some modifications. Briefly, biological triplicates of the WT, *pop1* and *pop6* cells were grown in YC media at 24 °C or 30 °C to $OD_{600} = 0.5$. Samples were treated with 750 μls of DMS for 10 mins at 24 °C or 4 mins at 30 °C. Untreated controls were grown similarly but without DMS. Isoamyl alcohol/Beta-mercaptoethanol was added to both DMS treated and untreated samples and the RNA was extracted, DNase treated, and ribosomal RNA was removed using the Ribo-zero Gold rRNA removal kit (Illumina). Reverse transcription was done using the TGIRT-III (Ingex) enzyme with primers for TLC1 (DG365, DG351 and DG353), NME1 (DG355), or ASH1 (DG357) RNAs (See supplementary Table 3 for sequences of primers). After reverse transcription the RNA:DNA hybrids were incubated with RNase H to release the cDNA products as previously described[36].

The cDNA was then used as a template to add adapters by PCR using iProof HF Polymerase (BIO-RAD) with tiled primers for TLC1 DG348 + DG365, DG364 + DG351, DG352 + DG353; NME DG354 + DG355 (Supplementary Table 3). Each PCR reaction was done separately and pooled together for sequencing at a final sample concentration of 1 ng/μl. The libraries were then prepared from the pooled samples and sequenced using the HiSeq Rapid Cluster Kit v2 (Illumina) and HiSeq

(2×180nt, 180 cycles per end) at the Princeton University core facility. Datasets generated for DMS-MaPseq are available through http://www.ncbi.nlm.nih.gov/bioproject/548768.

**Analysis of DMS-MaPseq data**. Illumina sequence data were split by sample with dual barcodes using barcode splitter[58], allowing 1 mismatch per barcode. Each sample was amplicon-split using an in-house script (split_amplicons.tcsh). Adapters were trimmed with cutadapt (Galaxy version 1.16)[59], and forward and reverse reads were merged using FLASH (galaxy version 1.2.11.3)[59] with ± 10 bp overlap. Reads were checked for quality using FastQC (galaxy version 0.71)[60]. Pairs that did not merge were discarded. Merged reads were then mapped using BWA-MEM (Galaxy version 0.7.1.7.1)[61,62] with default options. Mutations were identified using an in-house FreeBayes script (version v1.2.0-2-g29c4002; freebayes_div_and_conq.tcsh)[63]. DMS reactivities and local coverages were then calculated using an in-house script (count_snps.tcsh). Finally, unique amplicon abundances were calculated using mergeSeqs.pl (version 2.8 - a part of the CFF suite)[64]. All analysis steps were initiated via either galaxy workflow or batchCommander (version 3.12).

DMS reactivities were calculated from mutation counts and local coverages in the treated and untreated samples as follows. Let $X_k$ and $C_k$ denote the mutation count and coverage at nucleotide $k$, respectively. Using superscripts '+' and '-' for the treated and untreated samples, respectively, we computed the reactivity at $k$ as previously described[39].

$$r_k = max\left(\frac{\frac{X_k^+}{C_k^+} - \frac{X_k^-}{C_k^-}}{1 - \frac{X_k^-}{C_k^-}}, 0\right).$$

Reactivities were normalized using the 2-8% approach after masking the reactivities for guanines and uracils (Low and Weeks, 2010). Next, we used dStruct to perform differential analysis on the normalized reactivities[41].

We compared WT to *pop1* at 24 °C, WT to *pop6* at 24 °C, WT to *pop1* at 30 °C and WT to *pop6* at 30 °C for the ASH1, NME1 and TLC1 RNAs. We searched for differential regions that were at least 5 nts in length, with *P*-value <0.05 and False discovery rate-adjusted *P*-value (*q*-value) <0.1. The FDR control was done separately for each pairwise comparison of conditions. In addition, we ensured a minimum effect size by requiring mean $\Delta d > 0.01$ for these regions, where $\Delta d$ is the difference between within-strain and between-strain dissimilarity scores per nucleotide. It quantifies the degree to which the variation between strains surpasses that of biological replicates for each strain.

**TMG immunoprecipitation**. Immunoprecipitations were done as previously described[65] with some modifications. Briefly, 60 μg of total RNA were incubated with Anti-2,2,7-Trimethylguanosine Mouse mAb (K121) Agarose Conjugate (Millipore) over-night at 4 °C, followed by five washes with NET-2 buffer (150 mM NaCl, 0.05% NP-40, 50 mM Tris-HCl, pH 7.4) The immunoprecipitates were DNase treated with Turbo DNA-free and analyzed by RT-qPCR as specified above. TLC1 levels were measured using primers DG293 and DG294 for TLC1 and DG52 and DG53 for *ACT1* (see Supplementary table 3 for sequences). Data were quantified by IP/Input and normalized to levels of *ACT1* mRNA.

**3′ RACE**. To determine the sequences of the 3′ ends of TLC1 RNA in WT and *pop* cells, we performed 3′ RACE as previously described[66] with some modifications. Briefly, cells were grown in YC media to OD$_{600}$=0.5 at 24 °C and 30 °C and RNA was extracted as described above. Total RNA (5 μg) was DNase treated (Invitrogen AM107) as described above and ribosomal RNA was depleted with Ribo-zero Gold rRNA removal kit (Illumina). The RNA was concentrated using RNA clean and concentrator columns (Ribo-zero) and eluted in 11 μls of RNase-free water. Then, 600 ng of rRNA-depleted RNA was ligated to 5 μM of Universal miRNA Cloning Linker (New England BioLabs) for 24 h at 25 °C. After RNA clean up with RNA clean and concentrator-5 (Zymo research), the cDNA was synthesized using 5 pmol of universal RT primer (UP1) with SuperScript III reverse transcriptase (Invitrogen). PCR amplification was carried out using 2 μls of cDNA, 5 μM of the RT universal primer and TLC1 internal primer DG389 (see Supplementary Table 3 for sequences) with iProof HF Supermix (Bio-Rad). 3 μls of PCR products were analyzed on a 2.5% agarose gel to visualize TLC1 transcripts followed by Select-a-size DNA Clean and concentrator (Zymo-research) to retain products ≤600 nt. The amplicon samples were quantified using Qubit 2.0 Fluorometer (Life Technologies, Carlsbad, CA, USA) and the DNA integrity was checked using Agilent TapeStation (Agilent Technologies, Palo Alto, CA, USA).

**3′ RACE DNA library preparation and illumina sequencing**. DNA library preparations, sequencing reactions, and initial bioinformatics analysis were conducted at GENEWIZ (South Plainfield, NJ, USA). DNA library preparation, clustering, and sequencing reagents were used throughout the process using NEBNext Ultra DNA Library Prep kit following the manufacturer's recommendations (Illumina, San Diego, CA, USA). Adapter-ligated DNA was indexed and enriched by limited cycle PCR. DNA libraries were validated using a DNA 1000 Chip on the Agilent 2100 Bioanalyzer (Agilent Technologies, Palo Alto, CA, USA), and quantified by

Qubit and real time PCR (Applied Biosystems, Carlsbad, CA, USA). The DNA libraries multiplexed in equal molar mass were loaded on the Illumina MiSeq instrument according to manufacturer's instructions. The samples were sequenced using a 2×250 paired-end (PE) configuration. Image analysis and base calling were conducted by the MiSeq Control Software (MCS) on the Illumina MiSeq instrument. 3′ RACE data is available through https://www.ncbi.nlm.nih.gov/sra/PRJNA548560.

**3′ RACE data analysis**. The raw Illumina reads were checked for sequencing adapters and quality via FastQC. The reads were trimmed of their sequencing adapters and nucleotides with poor quality using Trimmomatic v. 0.36. Raw sequence data (.bcl files) generated from Illumina MiSeq were converted into fastq files and de-multiplexed using Illumina bsl2fastq v. 2.17 program. The paired reads were merged (FLASH-Galaxy 1.2.11.3) and mapped (BWA-MEM-Galaxy 0.7.1.7.1) as described above. The 5′ and 3′ end adapters were trimmed with Cutadapt (Galaxy version 1.16) and the sequences abundances were computed using a custom Galaxy workflow (Mature_RNA_Abundance_Analysis_Helper_1_(calc_abund_all)).

**RNA FISH**. Two 50-nt DNA probes were generated (Fig. 9a), one for TLC1 and one for ITS1, an internal segment of the 35 s rRNA precursor RNA that is removed in the nucleolus. The TLC1 probe (GW1), which targeted a 26 nt region of TLC1, was labeled with Cy3 (green) at its 5′ end, and the ITS1 probe (GW3), which targeted a 30 nt region of ITS1 RNA, was labeled with Cy5 (red) at its 5′ end. Both probes were designed to adopt a hairpin structure with a 10-nt overhang according to mFOLD[67] (see Supplementary Table 3 for sequences of probes). Unlabeled control probes targeted nucleotides 389-414 on the TLC1 transcript (GW2) or nucleotides 70-99 in ITS1 (GW4). Probes were reconstituted in TE buffer and subjected to an annealing cycle with denaturation at 95 °C and annealing at 55 °C in one-minute steps for 30 cycles. Cells were grown overnight in 150 mL of SD. Complete or SD -ura media at either 24 °C or 30 °C to an OD$_{600}$ of 0.6. Cells were prepared for FISH as described previously[47] with a few modifications. Cells were fixed in 10% V/V formaldehyde at room temperature for 45 min and spheroplasted and permeabilized overnight in ethanol. Hybridization was performed overnight at 30 °C with 50 nM FISH probes in a buffer containing 10% W/V dextran sulfate, 2 mM vanadyl ribonucleoside complex, 2X SSC, 0.001% W/V *E.Coli* tRNA, 0.3 μM bovine serum albumin, RNase free water, and 10% V/V formamide. Cells were pelleted, resuspended in 1 mL of wash buffer containing 2X SSC, 10% V/V formamide, RNase free water, and 1 μL of 1 mg/mL DAPI, and incubated for 10 minutes at 30 °C to wash out weakly-bound probes. Cells were washed once more without incubation and then aspirated and kept at 4 °C until imaged. 3 μL of concentrated cells were applied to a slide pre-cleaned with ethanol and mixed by gentle pipetting with 3 μL of oxygen scavenging buffer containing 10 mM Tris at pH 8, 2X SSC, 2.5 mM protocatechuic acid (PCA), 10Nm protocatechuate-3,4-dioxygenase (PCD) and 1 mM 6-hydroxy-2,5,7,8-tetramethylchroman-2-carboxylic acid (Trolox). Slides were sealed with epoxy. Z-stack images were acquired at 100 ms exposure by Micromanager[68], in 200 nm steps over 10 μm using a motorized stage. Laser output was set to produce 25 mW of power at the sample plane for both 640 nm and 532 nm illumination (1185055, Coherent; LCX-532L-100, Oxxius), and the LED controller was set to 0.5 V for UV illumination (M375L4, Thorlabs). The laser light was spun by a 2D-galvo system to achieve a uniform illumination[69]. Cell segmentation and FISH spot identification were performed using Matlab image processing toolbox. Nucleolus and nucleus locations were determined by intensity-based thresholding. Objects of volume smaller than 10 pixels were removed from the analysis. Spot location was extracted by applying a CLEAN algorithm[70]. In this algorithm, a Gaussian profile fit to a maximum intensity spot is sequentially subtracted from the original image until the intensity in the cell falls below the minimum acceptable intensity of a single fluorophore.

**Reporting summary**. Further information on research design is available in the Nature Research Reporting Summary linked to this article.

## Data availability

The 3′ RACE and DMS-MaPseq datasets generated and analyzed during the current study are available in the PRJNA548560 and PRJNA548768 repositories, respectively. The source data underlying Figs. 2c–g, 4a–f, 5a–d, 6c–d, 7, 8a, d, 9b–c and Supplementary Figs. 3–6 and 8–13 are provided as a Source Data file. All data is available from the corresponding author upon reasonable request.

## Code availability

Software for the analysis of 3′ RACE and DMS-MaPseq data is available at https://github.com/hepcat72/RNAseP-MRP/ and described in Methods

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

## Acknowledgements

We thank C. Boone and M. Costanzo for *pop* strains, J. Lucks for advice on structural analysis, S. Rouskin for advice on DMS-MaPseq, C. DeCoste and the Princeton Molecular Biology FACS facility for help with FACS and W. Wang and the Genomics Core Facility for help with sequencing. We thank C. Greider for suggesting the experiment on Yku titration. We are particularly grateful to A. Korennykh for his careful reading of the manuscript and his many valuable suggestions on the DMS protection experiments and their interpretation. Work in the Zakian lab is supported by 1R35GM118279 from the NIH. GMW and HDK acknowledge support from Georgia Institute of Technology startup funds and NIH grant R01-GM112882.

## Author contributions

Overall design and interpretation of experiments were done by P.D.G. and V.A.Z. R.W.L. processed deep sequencing data and designed code for the analysis of 3′ RACE and DMS-MaPseq. P.D.G. conducted all of the experiments except the localization of TLC1 in Fig. 9, which was conducted by G.M.W. with the guidance of H.K. G.M.W. also developed a custom analysis code for FISH localization. K.C. and S.A. performed differential reactivity analysis of the DMS-MaPseq. H.L. contributed to determining methods for analyzing the DMS-MaPseq data. P.D.G. and V.A.Z. wrote the paper with contributions from all authors.

## Conflict of interest

The authors declare no conflict of interest.
