## [Peer Review File · Nature Communications]

Reviewers' comments:

Reviewer #1 (Remarks to the Author):

In this manuscript, Garcia et al. report the telomerase-related phenotypes of pop mutants. This is an exciting topic in the field, since the Zakian, and later also Wellinger, group found RNase P/MRP Pop proteins (at least Pop1/6/7) associated with the telomerase RNP by mass spec. Here, Garcia et al. use temperature-sensitive pop strains (from Costanzo and Boone) to study effects of reduced pop function on telomerase. They look at telomerase protein subunit abundance and telomerase RNA localization, discovering that shorter telomeres at permissive/semipermissive temperatures are likely due to TLC1 RNA remaining largely cytoplasmic in the absence of full pop1/6 functionality. They also study TLC1 structure (and others), via collaboration, concluding that only the TLC1 internal loop where Pop proteins bind appears to be affected in the pop-deficient conditions.

Overall, there are a lot of very nice new data provided by this work and the manuscript comprises strong progress on the question of Pop proteins' function in telomerase. However, some results were expected based on very similar results from Lemieux et al. (Cell, 2016), such as the fact that mutations in the Pop-binding region of telomerase RNA led to reduced Est1 and Est2 association with the RNP. Furthermore, despite all of the data, the manuscript does not provide a much clearer idea as to why or how the Pop proteins are doing what the authors propose, such as promoting TLC1 nuclear localization and Est1/2 association with the RNA. The lack of a model explaining the authors' newly reported Pop proteins' functions in telomerase RNP biogenesis lessens my enthusiasm for the otherwise substantial collection of high-quality experimental results.

Major

The authors appear to be concluding that Pop proteins are primarily TLC1 nuclear-import factors, and that this is somehow related to Est1 and Est2 associating with the RNA. However, if one focuses on the amount of TLC1 in the nucleus as opposed to the fraction of molecules in the cytoplasm, there is even more TLC1 in the nucleus in pop1 and pop6 mutants than in wild type (Figure 9C). So how much influence on telomere length maintenance can/does the cytoplasmically retained TLC1 pool really have?

Why and how do the Pop proteins help TLC1 get to the nucleus? Having a strong sense of this seems very important to answer for the overall impact of this research. Furthermore, Est1, Est2, Est3, and Ku have all been reported to be required for TLC1 nuclear localization (Gallardo et al., EMBO 2008), which seems to be a serious conundrum in the field. Clearly these telomerase core and holoenzyme subunits' functions are not considered to be primarily localizing TLC1 to the nucleoplasm. Also, if Lemieux et al. are correct that Pop1/6/7 are directly important for telomerase activity (for example the modest improvement in an in-vitro assay in their 2016 Cell paper), how does this compare with the conclusion herein that the proteins simply allow TLC1 to get fully to the nucleus?

The claim of Pop-dependence of Est1 and Est2 association with TLC1 is part of Lemieux et al, Cell 2016,

based on mutations in TLC1 as opposed to the proteins. This lessens the novelty of the current result and the authors do not discuss this prior report sufficiently frankly in the manuscript as it is currently written.

To evaluate TLC1 secondary structure, the authors do an innovative DMS/deep-sequencing based analysis of various amplicons of TLC1 RNA, but the raw data (1) do not show a visually perceptible difference in the computationally derived RNA regions of difference (Figure 6D) and (2) cannot be seen at the resolution of individual nucleotides for any region. The authors must show the quantitated data for each condition at individual-nucleotide resolution across each amplicon at least in the Supplementary Data, and for the regions of significant difference in the main figures.

Finally, what is the nature of the pop mutations that make them TS? This is not explained, despite the paramount importance of the TS alleles and their phenotypes for this manuscript.

Minor

Page 12, line 253: the authors state that the (~30%) reduced telomerase assembly and binding of the Est2 subunit of the telomerase RNP "are sufficient to explain the short telomere phenotype of pop1 and pop6 cells." Without more quantitative explanation as to how the authors can conclude the telomere shortening can be explained by this degree of reductions in telomerase, this is an overstatement.

Figure 1B seems like it should also include telomerase acting at telomeres. Also, it seems as if, as drawn, after import of the fully mature RNP, it feeds back on its own transcription in step 1.

In Figure 1A, what is the evidence for Est1/3/2 physically bridging from the tip of the upper region of TLC1 to the catalytic center? This is unlikely given the Est1-site repositioning results supporting the flexible scaffold model in Zappulla and Cech (PNAS, 2004) and the stiff-armed TLC1 alleles in Lebo et al. (RNA, 2012).

Furthermore, on line 91 of page 5, the authors state that TLC1 is "highly structured" like RPR1 and NME1, but this conflicts with other statements about TLC1 being highly flexible, so the authors should adjust their wording to reflect what has been demonstrated.

There are arbitrary changes in colors used in the figure panels (the color does not track consistently with pop 1 vs. 6 or 24° vs. 30° etc), which does not help the reader quickly interpret the figures.

I really appreciate the methodological details in the manuscript, but some of it could be shifted to the Supplementary Information.

The references section needs editing, such as refs 22, 23, and 24.

Reviewer #2 (Remarks to the Author):

The authors investigated the function of RNase P/MRP in telomerase biogenesis. The component of RNase P/MRP, Pop1, Pop6 and Pop7 were previously found to bind directly the telomerase TLC1 from Wellinger lab (Lemieux et al, 2016 doi:10.1016/j.cell.2016.04.018; Laterreur et al. 2018 doi: 10.1261/rna.066696.118). They showed that removal of Pop binding region within TLC1 results in telomere shortening due to impaired Est1 binding. However, function of Pop protein in telomerase biogenesis was not investigated. Using temperature sensitive allele mutants of pop1 and pop6, the authors found that impaired stability of Pop1 and Pop6 results in impaired interaction of Est1 with TLC1 but maturation process and stability of TLC1 was not impaired. Cytological analysis showed that TLC1 accumulates at the cytoplasm in pop1 and pop6 mutants.

In my view, requirement of Pop1/6/7 binding to TLC1 for Est1 interaction was already demonstrated in a different way. Accumulation of TLC1 in the cytoplasm in the absence of Est1 is known (Gallardo et al 2008 doi: 10.1038/emboj.2008.21). The authors highlighted different function of Pop1/6/7 in TLC1 comparing to RNA components of RNase P/MRP in this manuscript. However, I did not find any data explaining how the differences in RNA folding were achieved and how Est1 interaction was impaired by the loss of Pop proteins.

Comments:

Overall manuscript was tedious and difficult to follow. Some paragraphs, especially in the result section, were fragmented. Many sentences are too complicated. I found difficult to read through. Each paragraph should contain conclusive message (aim, study, conclusion). Complex sentences should be separated and simplify. Result part is wordy. Most of subsections especially negative results should be concise in order to highlight positive results, ie new finding – the role of Pop1/6 in telomerase biogenesis. Discussion is almost the repeat of result. I wished to read more constructive and prospective messages. I like to read how new finding is significant and could be used or explain previously unsolved questions. The authors should compare with previous study about Pop1/6/7 in TLC1 in a constructive manner. They could highlight the difference in the phenotype of pop1 and pop6 mutants and discuss potentially additional role of Pop6? The major finding of this project is I think revealing the distinct role of Pop1/6 proteins in TLC1 and the RNA components of RNase P/MRP. How this difference was achieved should be discussed.

Some small things

1. Page 11 line 231, Fig 5B: The graph shows that Est2 binding is not significantly different in wt and pop1. This data cannot be interpreted as 'reduced'.
2. Western blot of Est1 and Est2 in Figure 4 looked different from these in Figure 5, where Est1

expression looked low in pop1.6 and Est2 looked normal.

3. Page 4 second paragraph is confusing. Information of Est1 and Yku may be presented separately.

4. The 2nd sentence of abstract and discussion: This is not the first paper reporting interaction of Pop1/6/7 with TLC1. This is misleading to the readers.

5. Page 7 line 131, all three mutants, but there is only two mutant strains pop1 and pop6 (and one wild type control) in Fig 2A and sup 1B

6. Whereas overall protein expression was not impaired in pop1 and pop6 mutants at permissive temperature, expression of Est2 and Pop1 was impaired. Do the authors have any clue to explain this phenomenon or speculate the possible reasons? Given pop proteins are RNA binding, the authors may check expression of these mRNA.

7. Figure 4C: Est2 expression was lower at 24°C but higher at 30°C in pop6 mutants compared to pop1. It looked even higher than wt when normalised to tubulin. Is this reproducible? The authors may show the graph of Est1 and Est2 at 30°C like Fig 4BD.

Reviewer #3 (Remarks to the Author):

Garcia et al study the effect of pop1 and pop6 mutants on yeast telomerase function. They find that telomeres are shortened. Even though TLC1 mRNA is more abundant, it is less associated with Est1 and Est2. This may be caused by improper localization of TLC1, which appears to localize more to the cytoplasm. Overall, this results in less holoenzyme at the telomeres.

In the proposed model, the mislocalization of TLC1 is key to the story. Although the localization changes appear significant, the smFISH protocol is deviating at several points from other published protocols. Despite these differences with previous protocols, the authors provide little control experiments and show only very limited raw data, making it hard to judge how reliable and specific this new protocol is.

The authors adapt a previous protocol from ref 42. Compared to this original paper, they use formaldehyde fixation instead of methanol fixation. For localization experiments, formaldehyde fixation is indeed preferred, but based on ref 42, this comes with the disadvantage of more non-specific spots with lower signal to noise. Perhaps this is fixed by the addition of a hairpin loop, but no control experiments are provided that show that the detected signal is indeed specific.

The authors mention that specificity of the probes is tested by the disappearance of signal when unlabeled probes were added. However, this does not test for specificity, since non-specific hybridization will also disappear when unlabeled probes are added. The best way to test for specificity is to test the probes on a TLC-deleted strain, which should result in loss of all spots.

In addition, the authors cite reference 40 cited to show that hairpin loops have a higher specificity, but this reference cannot be found in pubmed or on google scholar. The authors should either provide data supporting this claim, or cite a published and available paper.

For accurate counting measurements, each detected spot should represent a single RNA, and not multiple colocalized RNAs. The authors mention that the probes bleach in a single step (although the data is not shown), supporting this claim. Another way to test this, is to show that the histogram of spot intensities form a unimodal narrow distribution. Is this indeed the case? And do the distribution of intensities stay the same across conditions?

The authors claim that they detect roughly 50% of the total TLC1 transcripts. This is based on ref 42, but the experiments in ref 42 was performed in very different conditions:

(1) this paper uses hairpin probes labeled at the 5' end, whereas ref 42 uses on non-hairpin probes labeled internally; (2) cells are fixed in PFA instead of methanol; (3) the probe sequence and target RNA are completely different. Given that every probe sequence has its own efficiency, such an extrapolation from the data of ref 42 will be highly inaccurate. The authors should either test their efficiency or remove this statement from the paper.

Lastly, the authors provide one image of a wildtype cell and the smFISH experiment. To be able to judge the quantification and the claimed movement of the RNA to the cytoplasm, the authors should include some raw images from both wildtype and pop1 and pop6 mutants in one of the conditions where they see more cytoplasmic localization.

November 4, 2019

Dear Reviewers,

We thank the three reviewers for their thoughtful reviews of our paper. We addressed each comment from each reviewer. We feel that responding to the reviews has improved the paper enormously. The revised paper has been completely rewritten and reorganized. In addition, as indicated in specific comments, we added new data that were requested by reviewers. Another important change is our extending the interpretation of data from the RNA-IP and DMS-MaPseq experiments. This change was inspired by conversations with Dr. A. Korennykh, an RNA structure expert. He told us there was no way that the binding sites for Est1 (and other proteins) would have WT DMS-sensitivity unless the proteins were TLC1-bound at normal or near normal levels. His comments allowed us to realize that our data distinguish a role for Pop proteins in promoting Est1 binding to TLC1 from their acting to stabilize Est1-TLC1 interactions. Specifically, the DMS-MaPseq experiment, which is an *in vivo* assay, indicates that Est1 and Est2 bind to TLC1 at near WT levels in 24°C grown *pop* cells. However, by RNA-IP, Est1 (and to a much lesser extent Est2) binding to TLC1 is low at 24°C (14% of WT for Est1). Taken together, these data argue that Pop proteins are needed for the stability of the Est-TLC1 interactions, not for their binding. These ideas are incorporated throughout the revised paper. As a result of this change in interpretation, we also changed the order of presentation of some experiments.

Here we respond to each comment by each of the reviewers.

Reply to Reviewer 1:

"In this manuscript, Garcia et al. report the telomerase-related phenotypes of pop mutants. This is an exciting topic in the field, since the Zakian, and later also Wellinger, group found RNase P/MRP Pop proteins (at least Pop1/6/7) associated with the telomerase RNP by mass spec. Here, Garcia et al. use temperature-sensitive pop strains (from Costanzo and Boone) to study effects of reduced pop function on telomerase. They look at telomerase protein subunit abundance and telomerase RNA localization, discovering that shorter telomeres at permissive/semipermissive temperatures are likely due to TLC1 RNA remaining largely cytoplasmic in the absence of full pop1/6 functionality. They also study TLC1 structure (and others), via collaboration, concluding that only the TLC1 internal loop where Pop proteins bind appears to be affected in the pop-deficient conditions.

Overall, there are a lot of very nice new data provided by this work and the manuscript comprises strong progress on the question of Pop proteins' function in telomerase. However, some results were expected based on very similar results from Lemieux et al. (Cell, 2016), such as the fact that mutations in the Pop-binding region of telomerase RNA led to reduced Est1 and Est2 association with the RNP. Furthermore, despite all of the data, the manuscript does not provide a much clearer idea as to why or how the Pop proteins are doing what the authors propose, such as promoting TLC1 nuclear localization and Est1/2 association with

the RNA. The lack of a model explaining the authors' newly reported Pop proteins' functions in telomerase RNP biogenesis lessens my enthusiasm for the otherwise substantial collection of high-quality experimental results."

We thank the reviewer for his/her positive comments.

Major points (although the points are not numbered by the reviewer, we number them here for ease of discussion):

1. *"The authors appear to be concluding that Pop proteins are primarily TLC1 nuclear-import factors, and that this is somehow related to Est1 and Est2 associating with the RNA."* We do not propose that Pop proteins are TLC1 import factors. Rather, our model is that Pop proteins are important for the stable association of Est1 (and hence Est3) (and to a much lesser extent Est2) with TLC1. These associations occur in the cytoplasm. Our model proposes that in late S/G2 phase, the factors that import telomerase into the nucleus recognize feature(s) of the telomerase holoenzyme. Therefore, reduced abundance/activity of Pop proteins results in a lower probability of stable holoenzyme formation; i.e., there is less substrate with the correct structure for recognition by telomerase import factors. As a result, TLC1 accumulates in the cytoplasm. We apologize that our model was not stated more clearly in the original version of the paper. In addition to a more explicit verbal description of the model at the very end of the discussion, we now include a model figure (Fig. 10).

2. *"However, if one focuses on the amount of TLC1 in the nucleus as opposed to the fraction of molecules in the cytoplasm, there is even more TLC1 in the nucleus in pop1 and pop6 mutants than in wild type (Figure 9C). So how much influence on telomere length maintenance can/does the cytoplasmically retained TLC1 pool really have?"* This question gets at the fundamental conundrum of the data: *pop* cells have 2X as much TLC1 (thought to be the limiting component) and near WT levels of Est1/2 at 24°C, yet they have short telomeres. Because there is more TLC1 in *pop* than in WT cells, the telomere length defect cannot be explained by a cellular deficiency of TLC1. As the reviewer points out, there are ~2X more TLC1 molecules in the nucleoplasm (NP) in *pop1* versus WT cells (the amount of TLC1 in the NP is not significantly different in *pop6* vs WT). However, compared to WT, in *pop1* cells, there is 5X more cytoplasmic TLC1 at 24°C. The increase in cytoplasmic TLC1 is even greater in *pop6* cells at 24°C and in both *pop1* and *pop6* cells at 30°C. Although we present both the absolute number and the fractional number of TLC1 spots (Fig. 9D), because there is more TLC1 in *pop* mutants, we believe that the appropriate way to assess the distribution of TLC1 molecules is the fraction of TLC1 molecules in each cellular compartment. The interpretation that TLC1 accumulates in the cytoplasm in *pop* cells is supported by the greater amount of mature TLC1 in *pop* versus WT cells, as the nucleases that are thought to degrade TLC1 are not cytoplasmic.

3. *"Why and how do the Pop proteins help TLC1 get to the nucleus? Having a strong sense of this seems very important to answer for the overall impact of this research. Furthermore, Est1, Est2, Est3, and Ku have all been reported to be required for TLC1 nuclear localization (Gallardo et al., EMBO 2008), which seems to be a serious conundrum in the field. Clearly these telomerase core and holoenzyme subunits' functions are not considered to be primarily localizing TLC1 to the nucleoplasm."* As stated in point 1, we do not propose a role for Pop proteins in TLC1 import, but rather propose that they stabilize the substrate that is the target of the import factors. Our model that import factors recognize some aspect of the

holoenzyme also explains why nuclear localization of TLC1 is impaired in cells lacking Est proteins.

4. *“Also, if Lemieux et al. are correct that Pop1/6/7 are directly important for telomerase activity (for example the modest improvement in an in-vitro assay in their 2016 Cell paper), how does this compare with the conclusion herein that the proteins simply allow TLC1 to get fully to the nucleus?”* Again, this is not our model. However, our data do provide a plausible explanation for the stimulating effects of Pop proteins on the *in vitro* telomerase assay (we added this explanation in the revised manuscript in response to reviewer 1’s comment). It has always been perplexing that extracts prepared from *est1Δ* or *est3Δ* cells have the same telomerase activity as WT extracts. These data are even more perplexing once it became clear that Est1 is essential to activate telomerase *in vivo* (Chen et al., 2018). The strain used by Lemieux et al. to make the extract for the *in vitro* telomerase reaction contains all three Est proteins. Therefore, one explanation for the stimulating effects of adding Pop1 to an *in vitro* telomerase is that it stabilizes the association of Est1 and hence Est3 with telomerase. This stimulation should render the normally Est1-independent *in vitro* reaction Est1-sensitive. Perhaps, the Wellinger lab (or another group) will test this idea by adding Pop1 to a telomerase extract prepared from *est1Δ* cells.

5. *“The claim of Pop-dependence of Est1 and Est2 association with TLC1 is part of Lemieux et al, Cell 2016, based on mutations in TLC1 as opposed to the proteins. This lessens the novelty of the current result*

We are fans of the Lemieux et al 2016 paper. However, our paper addresses the consequences of reduced/mutant Pop proteins on telomeres and telomerase, an area not addressed at all in Lemieux et al. Here we list the findings that are unique to our paper.

- i. Pop proteins are required for maintenance of telomere length: telomeres are ~30% shorter than WT in *pop1* and *pop6* cells at 24°C (even shorter at 30°C).
- ii. Pop proteins are required for WT levels of TLC1; levels of TLC1 are 2x higher in *pop* cells compared to WT at 24°C (even higher at 30°). This result was completely unexpected as levels of NME1 and RPR1 are Pop protein-dependent.
- iii. Pop proteins have very different effects on NME1 and RPR1 versus TLC1. For example, while TLC1 is more abundant in *pop* cells, NME1 is less abundant (also see v. below). This result was also completely unexpected, given that the regions to which they bind in the three RNAs are interchangeable *in vivo* (Lemieux et al.,). We think this difference is one of the most interesting findings in our paper.
- iv. TLC1 is processed normally at both its 5’ and 3’ ends in *pop* cells. Although non-canonical targets of RNase P, have been described in bacteria and metazoans, in these cases the RNA is processed by the RNase activity of RNase P. By showing that Pop proteins are not required to process TLC1, we provide the first example, to our knowledge, of Pop proteins acting independently of an RNase complex. This finding opens up a whole new way of thinking about Pop proteins that might inspire others to consider that they affect “their” RNA by a mechanism other than promoting RNA processing.
- v. Pop proteins have been thought for many years to promote the folding of NME1 and RPR1. Here we show this directly for NME1, whose global structure is perturbed in *pop* cells even at 24°C. In contrast, the structure of TLC1 is largely unaffected. Thus, the

telomerase defects in *pop* cells are not due to misfolded TLC1. This is a key and totally unanticipated finding in our paper (also relevant to point iii above).

vi. Pop proteins are required for efficient holoenzyme binding to telomeres *in vivo*.

vii. Like Lemieux et al., we used RNA-IP to monitor Est1 and Est2 binding to TLC1. However, we also carried out TLC1 structure analysis by DMS sensitivity. The combination of the two methods allowed us to conclude that *pop* cells are not defective in Est1 or Est2 binding to TLC1, but rather the stability of this binding is impaired. Therefore, even in the area of greatest overlap, our paper adds an important mechanistic insight into how Pop proteins affect telomerase. Our paper also provides a paradigm for how to distinguish these two functions experimentally.

viii. Telomerase RNA is over-represented in the cytoplasm in *pop1* and *pop6* cells. We hypothesize that the import factors that escort the holoenzyme to the nucleus in late S/G2 phase, recognize some aspect of the holoenzyme. Owing to the instability of the Est1-TLC1 interactions, the holoenzyme is less abundant in *pop* cells, explaining the accumulation of TLC1 in the cytoplasm. These data provide an additional unanticipated consequence of reduced Pop function. Moreover, accumulation of TLC1 in the cytoplasm provides an explanation for elevated amounts of TLC1 in *pop* cells.

We hope the reviewer agrees that our paper provides multiple new insights that enhance and greatly extend the role of Pop proteins on telomerase presented in Lemieux et al. We feel both papers make distinct contributions to our understanding of how Pop proteins affect telomere biology.

6. *“and the authors do not discuss this prior report [that Est1 and Est2 are not associated with TLC1 in a mutant that lacks the P3-like region in Lemieux et al., 2016] sufficiently frankly in the manuscript as it is currently written.”*

We’re puzzled by this statement. We reported in both the introduction and the discussion that Lemieux et al. 2016 found reduced Est1 association with a mutant form of TLC1 that lacks the Pop binding site. The introduction said: “Although the Est1 and Pop binding sites are separable (Fig. 1), deletion of the CS2a/TeSS domain abolishes Est1 binding to TLC1 (Lemieux et al., 2016). Addition of Pop1 enhances telomerase activity *in vitro*, although the mechanism by which Pop proteins affect telomerase *in vivo* is not known nor is it known if Pop proteins affect telomerase abundance or telomere length”. In the discussion (lines 468-472), we wrote “The deficient association of Est1 and somewhat reduced Est2 to a deletion derivative of TLC1 that lacks the CS2a/TeSS domain led to the proposal that Pop proteins stabilize Est1 and to a lesser extent Est2 binding to TLC1 (Lemieux et al., 2016). However, the reduced Est binding to this deleted version of TLC1 could be due to disruption of its structure. Our data led to a similar conclusion based on the behavior of WT TLC1 and its entrapment in the cytoplasm when levels of Pop proteins are low.” In the revised manuscript, their result is also mentioned in the results section (“Using RNA-IP, Est1 (and to a lesser extent Est2) bind at reduced levels to a deletion derivative of TLC1 that lacks the P3-like CS2a domain to which Pop proteins bind (Lemieux et al., 2016).”; lines 264-266.) We think we did a good job discussing these data in the original version of the manuscript and hope that upon further reflection, reviewer 1 agrees. In the revised paper, there is even more discussion of these results in light of our ability to show that Est1 and Est2 bind TLC1 at WT levels in *pop* cells, but this binding is unstable. As Lemieux et al. determined Est1/2

binding only by RNA-IP whereas we used both DMS-protection and RNA-IP, they were unable to distinguish between reduced and unstable binding (lines 485-488).

7. *“To evaluate TLC1 secondary structure, the authors do an innovative DMS/deep-sequencing based analysis of various amplicons of TLC1 RNA, but the raw data (1) do not show a visually perceptible difference in the computationally derived RNA regions of difference (Figure 6D) and (2) cannot be seen at the resolution of individual nucleotides for any region. The authors must show the quantitated data for each condition at individual-nucleotide resolution across each amplicon at least in the Supplementary Data, and for the regions of significant difference in the main figures.”*

In the original version of the paper, we showed all the regions of NME1 and the one 7 nt stretch in TLC1 that were differentially DMS-sensitive in *pop* and WT cells at 24°C in figure 8 within the main text. The regions of both RNAs that were differentially DMS-sensitive at 30°C were shown in the Supplemental Figures. We changed the presentation of these figures in several ways. First, the differences in DMS-sensitivity are shown at nucleotide resolution. These differences are hopefully more evident in the revised figures (revised fig. 8, Supp Fig 6 and Supp Fig14). Second, we present the data for all the portions of NME1 and TLC1 that were analyzed in Supplementary Figures 4-14. However, presenting all of the data even in Supplementary figures at nucleotide resolution is a huge amount of information as the experiments are done \pm DMS, at two different temperatures (24° and 30°), in three different strains (WT, *pop1*, *pop6*), and on three RNAs. Therefore, we do not present the data for the negative control ASH1 (The structured end of ASH1 as probed by DMS-MaPseq in WT cells is published, and our data agree completely with the earlier analysis.)

8. *“Finally, what is the nature of the pop mutations that make them TS?”* We added these new data to the text of the revised manuscript as well as in revised Fig. 2A (*pop1*) and B (*pop6*).

Minor points made by reviewer 1.

1. *“Page 12, line 253: the authors state that the (~30%) reduced telomerase assembly and binding of the Est2 subunit of the telomerase RNP “are sufficient to explain the short telomere phenotype of pop1 and pop6 cells.” Without more quantitative explanation as to how the authors can conclude the telomere shortening can be explained by this degree of reductions in telomerase, this is an overstatement.”*

We softened the statement on page 12 (which is now on page 14, line 289-90) to “Impaired stability of the telomerase holoenzyme and its reduced telomere binding likely explain the short telomere phenotypes of *pop1* and *pop6* cells at 24°C.”

2. *“Figure 1B seems like it should also include telomerase acting at telomeres. Also, it seems as if, as drawn, after import of the fully mature RNP, it feeds back on its own transcription in step 1.”*

We thank the reviewer for pointing out that the original version of figure 1B appeared to suggest that the mature RNP affects its own transcription. We changed Fig. 1B to show telomerase acting on telomeres in a way that cannot be confused with effects on transcription.

3. *“In Figure 1A, what is the evidence for Est1/3/2 physically bridging from the tip of the upper region of TLC1 to the catalytic center? This is unlikely given the Est1-site repositioning results supporting the flexible scaffold model in Zappulla and Cech (PNAS, 2004) and the stiff-armed TLC1 alleles in Lebo et al. (RNA, 2012).”*

The reviewer objected to Figure 1A which shows Est1/3/2 physically bridging from the tip of the upper region of TLC1 to the catalytic center. Fig. 1A is clearly not an atomic resolution view but more of a cartoon that is meant to illustrate that Est1 binds to the Est1 arm of TLC1, Est3 binds directly to both Est1 and Est2, and Est2 binds to the catalytic center of TLC1. We clarify in the figure legend that this is static view of the holoenzyme that is meant to illustrate the above points.

4. *"On line 91 of page 5, the authors state that TLC1 is "highly structured" like RPR1 and NME1, but this conflicts with other statements about TLC1 being highly flexible, so the authors should adjust their wording to reflect what has been demonstrated."* We're not sure we completely understood the reviewer's point. To us, highly structured refers to the long base paired arms emanating from the central core of the RNA. Flexible referred to data from the Zapulla lab that the functions of the arms don't require a single, specific arrangement. We deleted the word flexible and hope that this deletion satisfies the reviewer's objection.
5. *"There are arbitrary changes in colors used in the figure panels (the color does not track consistently with pop 1 vs. 6 or 24° vs. 30° etc), which does not help the reader quickly interpret the figures."* As suggested, we made the colors for strains and temperatures consistent from figure to figure.
6. *"I really appreciate the methodological details in the manuscript, but some of it could be shifted to the Supplementary Information."* We kept this statement in mind as well as the comments from reviewer 2 on our writing style in revising the paper.
7. *"The references section needs editing, such as refs 22, 23, and 24."* Done (thanks for catching these errors.)

Reply to Reviewer 2:

1. *"Overall manuscript was tedious and difficult to follow. Some paragraphs, especially in the result section, were fragmented. Many sentences are too complicated. I found difficult to read through. Each paragraph should contain conclusive message (aim, study, conclusion). Complex sentences should be separated and simplify. Result part is wordy. Most of subsections especially negative results should be concise in order to highlight positive results, ie new finding – the role of Pop1/6 in telomerase biogenesis. Discussion is almost the repeat of result. I wished to read more constructive and prospective messages. I like to read how new finding is significant and could be used or explain previously unsolved questions. The authors should compare with previous study about Pop1/6/7 in TLC1 in a constructive manner."* We revised and reorganized the entire manuscript with the reviewer's specific criticisms on style in mind. We hope that the reviewer finds the revised version easier to read.
2. *"They could highlight the difference in the phenotype of pop1 and pop6 mutants and discuss potentially additional role of Pop6? The major finding of this project is I think revealing the distinct role of Pop1/6 proteins in TLC1 and the RNA components of RNase P/MRP. How this difference was achieved should be discussed."* In most assays, pop6 cells have a more severe phenotype than pop1 cells. However, we don't think it is appropriate to argue from these data that Pop6 has additional functions. It is known from earlier work (cited in the paper) that a Pop6/7 heterodimer binds to the P3 region and recruits Pop1. We show that the mutant Pop1 protein is less abundant in pop6 cells than in pop1 cells. We suspect that mutant Pop1 is more

likely to be degraded when its binding to the P3-like region is reduced owing to mutant *pop6*. However, we consider these ideas as being too speculative to suggest them in this paper. We discuss why NME1 is less abundant in *pop* cells while TLC1 is more abundant. Specifically, the global structure of NME1, but not TLC1, is perturbed in *pop* cells. We propose this perturbed structure targets NME1 for degradation. In contrast, TLC1 is over-represented in the cytoplasm, which sequesters it from the nuclear nucleases that degrade it.

3. *"In my view, requirement of Pop1/6/7 binding to TLC1 for Est1 interaction was already demonstrated in a different way."* This concern is similar to point 7 from reviewer 1. As noted in our response to reviewer 1, our paper provides far more information than the demonstration that Pop proteins affect the stability of the Est-TLC1 interactions. Most of our paper is non-overlapping with the earlier Lemieux et al. paper as we focus on the consequences of reduced/mutant Pop proteins on telomeres and telomerase, an area not addressed at all in Lemieux et al. Here we repeat our comments to reviewer 1 of the findings that are unique to our paper.
 - i. Pop proteins are required for maintenance of telomere length: telomeres are 30% shorter than WT in *pop1* and *pop6* cells at 24°C (even shorter at 30°C).
 - ii. Pop proteins are required for WT levels of TLC1; levels of TLC1 are 2x higher in *pop* cells compared to WT at 24°C (even higher at 30°C). This result was completely unexpected as levels of NME1 and RPR1 are Pop protein-dependent.
 - iii. Pop proteins have very different effects on NME1 and RPR1 versus TLC1. For example, while TLC1 is more abundant in *pop* cells, NME1 is less abundant (also see *v* below). This result was completely unexpected, given that the regions to which they bind in the three RNAs are interchangeable *in vivo* (Lemieux et al.,). We think this difference is one of the most interesting findings in our paper.
 - iv. TLC1 is processed normally at both its 5' and 3' ends in *pop* cells. Although non-canonical targets of RNase P, have been described in bacteria and metazoans, in these cases the RNA is processed by the RNase activity of RNase P. By showing that Pop proteins are not required to process TLC1, we provide the first example, to our knowledge, of Pop proteins acting independently of an RNase complex. This finding opens up a whole new way of thinking about Pop proteins that might inspire others to consider that they affect "their" RNA by a mechanism other than promoting RNA processing.
 - v. Pop proteins have been thought for many years to promote the folding of NME1 and RPR1. Here we show this directly for NME1, whose global structure is perturbed in *pop* cells even at 24°C. In contrast, the structure of TLC1 is largely unaffected. Thus, the telomerase defects in *pop* cells are not due to misfolded TLC1. This is a key and totally unanticipated finding in our paper (also relevant to point iii above).
 - vi. Pop proteins are required for efficient holoenzyme binding to telomeres *in vivo*.
 - vii. Like Lemieux et al., we used RNA-IP to monitor Est1 and Est2 binding to TLC1. However, we also carried out TLC1 structure analysis by DMS sensitivity. The combination of the two methods allowed us to conclude that *pop* cells are not defective in Est1 or Est2 binding to TLC1, but rather the stability of this binding is impaired. Therefore, even in the area of greatest overlap, our paper adds an

important mechanistic insight into how Pop proteins affect telomerase. It also provides a paradigm for how to distinguish these two functions experimentally.

viii. Telomerase RNA is over-represented in the cytoplasm in *pop1* and *pop6* cells. We hypothesize that the import factors that escort the holoenzyme to the nucleus in late S/G2 phase, recognize some aspect of the holoenzyme. Owing to the instability of the Est1-TLC1 interactions, the holoenzyme is less abundant in *pop* cells, explaining the accumulation of TLC1 in the cytoplasm. These data provide a totally novel and unanticipated consequence of reduced Pop function. In addition, accumulation of TLC1 in the cytoplasm provides an explanation for elevated amounts of TLC1 in *pop* cells.

We hope the reviewer agrees that our paper provides multiple new insights that enhance and greatly extend the role of Pop proteins on telomerase presented in Lemieux et al. We feel both papers make distinct contributions to our understanding of how Pop proteins affect telomere biology.

4. Accumulation of TLC1 in the cytoplasm in the absence of Est1 is known (Gallardo et al 2008 doi: 10.1038/emboj.2008.21)."

Gallardo et al. 2008 shows reduced nuclear TLC1 (~30% WT levels) in the absence of Yku or any of the three Est proteins. In contrast, we show that reduced Pop proteins results in cytoplasmic accumulation of TLC1 in cells with WT levels of Est1, Est2, and, presumably, YKu. This is a very different result than saying the absence of Est proteins traps TLC1 in the cytoplasm. The over-representation of TLC1 in the cytoplasm provides a satisfying explanation for why TLC1 is more abundant in *pop* cells.

4. "I did not find any data explaining how the differences in RNA folding were achieved and how Est1 interaction was impaired by the loss of Pop proteins." We hope that the reviewer will find that the revised manuscript provides a better explanation for these points. To summarize, the DMS-MaPseq data show that only 7 nucleotides in TLC1 have altered DMS-sensitivity in *pop* cells (Fig. 6 D,E). These 7 nts define the site of Pop protein binding (as determined by Lemieux et al. 2016). As noted in both the original and revised version, altered sensitivity of these nucleotides is best explained by greater exposure of these nucleotides to DMS as a result of reduced Pop binding. This hypothesis is supported by our finding that Pop1 is present at only ~30% of WT levels in *pop1* and 5% of WT in *pop6* cells at 24°C. The Est1, Est2, Yku, and Sm binding sites are not perturbed at 24°C. Because these binding sites are not differentially sensitive to DMS in *pop* cells *in vivo* at 24°C, these proteins likely bind to TLC1 at WT levels at this temperature. By DMS-MaPseq, the Est1 site is perturbed at 30°C, which can be explained by the reduced levels of Est1 at this temperature. In contrast, NME1 structure in the same cells is globally perturbed in *pop* cells at 24°C (and also at 30°C). This global perturbation can be explained by the reduced binding of Pop1/6/7 (Pop6 and 7 bind as a heterodimer, which recruits Pop1.) Binding of most of the 8 other protein subunits of RNase MRP is dependent on the binding of Pop1/6/7. The RNA-IP experiments in our paper show that only ~14% of TLC1 molecules are stably Est1 associated at 24°C. As explained in the revised manuscript, in combination with the DMS data, this finding demonstrates that Est1-TLC1 binding is unstable in *pop* cells.

Small things:

1. *"Page 11 line 231, Fig 5B: The graph shows that Est2 binding is not significantly different in wt and pop1. This data cannot be interpreted as 'reduced'."*
In response to the reviewers' comment, we changed the statement to "We conclude that mutant Pop1 reduces the stable association of Est1 to TLC1 (and in *pop6* cells, Est2)."
2. *"Page 4 second paragraph is confusing. Information of Est1 and Yku may be presented separately."* As suggested, we separated the statements about Yku and Est1.
3. *"The 2nd sentence of abstract and discussion: This is not the first paper reporting interaction of Pop1/6/7 with TLC1. This is misleading to the readers."* The abstract reads: "RNase P and MRP are highly conserved, multi-protein/RNA complexes with essential roles in processing ribosomal and tRNAs. Three proteins found in both complexes, Pop1, Pop6, and Pop7 are also telomerase-associated. Here, we determine....". The first two sentences provide the background for the current paper. The third sentence, which begins "here we determine" indicates that from here on, we are talking about work presented in the current manuscript; i.e., the first two sentences are published information. Throughout our paper, including the abstract, we use standard science tense usage "rules" that help the reader distinguish between published work and data presented in the current manuscript. That is, published results are expressed in present tense while results in the current paper are in past tense (except for conclusions, which are in present tense.) The rules that we follow are stated in the guidelines for ASM papers.
4. *"Page 7 line 131, all three mutants, but there is only two mutant strains pop1 and pop6 (and one wild type control) in Fig 2A and sup 1B."*
We corrected the error.
5. *"Western blot of Est1 and Est2 in Figure 4 looked different from these in Figure 5, where Est1 expression looked low in pop1.6 and Est2 looked normal".* Quantitation of Est1 levels over three different westerns (only one of which is shown) shows that Est1 abundance at 24° is not statistically different in *pop1*, 6, and WT cells (Fig. 4B). In Figure 5, both input (equivalent to westerns in Fig 4) and IP westerns are shown; the loading control isn't shown as these westerns are not meant to make a quantitative statement about Est1 and Est2 levels but rather to show that Est1 and Est2 are immuno-precipitated as efficiently in *pop* as in WT cells. Note that input Est1 levels in Fig. 5C look the same in *pop* and WT cells.
6. *"Whereas overall protein expression was not impaired in pop1 and pop6 mutants at permissive temperature, expression of Est2 and Pop1 was impaired. Do the authors have any clue to explain this phenomenon or speculate the possible reasons. Giving pop proteins are RNA binding, the authors may check expression of these mRNA."* Many investigators find that temperature sensitive proteins are less abundant than WT proteins, even at permissive temperature. Thus, reduction of Pop1 is almost surely due to the seven amino acid changes in its sequence that confer its temperature sensitive phenotype (new data in Fig. 2A).
7. *Figure 4C: Est2 expression was lower at 24°C but higher at 30°C in pop6 mutants compared to pop1. It looked even higher than wt when normalised to tubulin. Is this reproducible?* Est2 levels are somewhat elevated in *pop1* cells and somewhat reduced in *pop6* cells. As the experiments were done on at least three biological

replicates, the data are reproducible. We don't think it's worth exploring the basis for these relatively small differences, especially as they are not in the same direction in the two mutants.

8. *The authors may show the graph of Est1 and Est2 at 30°C like Fig 4BD.* These data are now shown in expanded Figure 4 B and D.

Reply to Reviewer 3: All of this reviewer's comments are directed at the FISH experiments. The reviewer suggested several controls to improve the validity of the probes used in this analysis. We thank the reviewer for these suggestions.

1. *"The authors mention that specificity of the probes is tested by the disappearance of signal when unlabeled probes were added. However, this does not test for specificity, since non-specific hybridization will also disappear when unlabeled probes are added. The best way to test for specificity is to test the probes on a TLC-deleted strain, which should result in loss of all spots."* We thank the reviewer for this excellent suggestion. We provide new data that show very low signal when the TLC1 probe was tested on a *tlc1Δ* strain (0.61 ± 0.14 spots per cell) vs WT cells (14.69 ± 0.46 spots per cell) (Supplemental Fig. 17).

2. *"In addition, the authors cite reference 40 cited to show that hairpin loops have a higher specificity, but this reference cannot be found in pubmed or on google scholar. The authors should either provide data supporting this claim, or cite a published and available paper."* In Supplemental Fig. 15, we present data that *indirectly* support the higher specificity of hairpin probes compared to linear probes (originally published in the PhD thesis of Gable Wadsworth). Because all of the bases on a linear probe are exposed, they can base-pair with complementary bases on an RNA molecule independently of each other. Therefore, a linear probe is expected to stably hybridize to an RNA molecule containing only a few mismatched bases. On the other hand, the exposed bases in a hairpin probe are limited to the toehold region (Supplemental Fig. 15A). Stable hybridization of the hairpin probe to its target RNA thus requires the unzipping of the hairpin stem, one base at a time. In support of this idea, we show that even a single mismatch in the target can dramatically inhibit stem unzipping (Supplemental Fig. 15D). Therefore, the hairpin design is expected to increase probe specificity by adding sequential kinetic checkpoints in the annealing pathway, similar to the kinetic proofreading mechanism. A recent paper also notes that a linear probe is less specific than a hairpin probe (Marras, et al. 2019). We added this reference to the revised paper.

3. *"For accurate counting measurements, each detected spot should represent a single RNA, and not multiple colocalized RNAs. The authors mention that the probes bleach in a single step (although the data is not shown), supporting this claim. Another way to test this, is to show that the histogram of spot intensities form a unimodal narrow distribution. Is this indeed the case? And do the distribution of intensities stay the same across conditions?"*

Following the referee's suggestion, we provide new data in Supplemental figure 18. In A, we present the distribution of FISH spot intensities, which show one dominant Gaussian distribution with a tail at higher intensity values. Fitting this distribution to a Gaussian mixture model yields 85% and 15% for the low- and high-intensity populations, respectively. The photobleaching test (B and C) shows that the majority of the FISH spots photobleach in a single step (84%) vs. two steps (16%) during the 5-minute acquisition

period. These two different methods are both consistent with the interpretation that the majority of spots (~85%) represent a single RNA molecule.

4. *“The authors claim that they detect roughly 50% of the total TLC1 transcripts. This is based on ref 42, but the experiments in ref 42 was performed in very different conditions:*

(1) this paper uses hairpin probes labeled at the 5' end, whereas ref 42 uses on non-hairpin probes labeled internally; (2) cells are fixed in PFA instead of methanol; (3) the probe sequence and target RNA are completely different. Given that every probe sequence has its own efficiency, such an extrapolation from the data of ref 42 will be highly inaccurate. The authors should either test their efficiency or remove this statement from the paper.”

We agree with this referee that ~50% detection efficiency cannot be directly applied to TLC1 FISH due to differences in labeling scheme, fixative, and sequence. Therefore, we point out these differences, but note that if we extrapolate from the earlier study, our numbers are in line with earlier measurements of TLC1 copy number. [p.18-9, line 393-402: “Using a probe for a different sequence and some methodological differences in the labeling scheme and fixative, the single-probe FISH protocol detected roughly 50% of cellular transcripts (Wadsworth et al., 2017). If we extrapolate from these earlier experiments, the ~11 Cy3-labeled spots in WT cells at 30°C predicts ~22 TLC1 molecules per WT cell. This estimate is in good agreement with a previous estimate of 29 TLC1 molecules per cell as determined by RT-PCR (Mozdy and Cech, 2006)”. Although this is not exactly what the reviewer requested, we hope this change satisfies the reviewer’s concerns. We also present the effects of different fixatives on FISH images in Supplemental Figure 16.

5. *“Lastly, the authors provide one image of a wildtype cell and the smFISH experiment. To be able to judge the quantification and the claimed movement of the RNA to the cytoplasm, the authors should include some raw images from both wildtype and pop1 and pop6 mutants in one of the conditions where they see more cytoplasmic localization.*

These data are now shown in Supplemental Fig 19. We note that the spots appearing in the 2D projection of the nucleus may or may not be in the nucleus in 3D space. We also note that TLC1 does not in general appear diffuse throughout the cell but tends to be near the nucleus (within approximately 20 pixels of the boundary assigned to the nucleus).

We hope that you and the reviewers find that our revised manuscript is now suitable for publication in *Nature Communications*.

Best regards,

Reviewers' comments:

Reviewer #1 (Remarks to the Author):

Ever since the Zakian and Wellinger labs published (in 2015 and 2016) that Pop proteins bind not only to RNase MRP and P RNAs, but also to telomerase RNA, TLC1, the obvious question in the field has been: why do these proteins also associate with telomerase RNA? On a related note, Garcia et al. state that “the goal of this paper is to determine the functional significance of the association of Pop proteins with telomerase.” As I stated in my original review, there are a lot of thoroughly presented data in this manuscript. Certainly, these data provide a foundation for studying Pop1 and Pop6 TS alleles and make some headway on understanding what Pop1 and Pop6 do and do not do. But there are limitations to how much has, or can, be learned from partial loss-of-function phenotypes of TS mutants studied at permissive and semi-permissive temperatures.

Garcia et al. claim that the Pop1 and Pop6 proteins bind to TLC1 to help Est1, and to a lesser extent Est2, stay associated with TLC1, and to help certain pools of telomerase RNP to fully migrate to the nucleus. Comparing the take-home messages with what was published in Lemieux et al. Cell 2016, this Est1/Est2 association conclusion is not novel, even if it was more carefully performed here. And fundamental questions remain unresolved, such as whether Pop proteins have direct roles in catalysis, and other hypotheses proposed by the Wellinger group. Lemieux et al. 2016 had concluded that Pop1/6/7 was having a vast, critical influence on telomerase RNP folding and assembly, and Garcia et al. make clear progress on this question by showing with in vivo DMS probing that only <1% of the analyzed nucleotides in TLC1 were differentially reactive in pop cells compared to 26% in NME1. This demonstrates that Pop proteins only influence TLC1 within the 7-nt position to which they directly bind. This sort of progress synthesizes the impact of Garcia et al.’s exhaustive data. By digging deeper into the claims of Lemieux et al., it seems that the Pop1/6/7 complex, as is so often the case with telomerase RNP across species from ciliates to humans, has evolved to coopt a protein complex from a different RNP to promote its own optimal biogenesis. This story is not, however, what is stated by Garcia et al. in its currently written form, lessening the impact of the work. Because of the current Figure 10 model and the overall collection of negative and partial phenotypes, it is questionable as to whether this manuscript warrants publication in a top journal such as Nature Communications. But I think that the impact is greater if the authors revisit how they view the impact of their findings by showing how they refine where Lemieux et al. concluded and by modifying Figure 10 (see below).

In responding to the reviewers, Garcia et al. have rephrased much of the manuscript, reordered some figures, and, importantly, added a final figure showing their model. It was great that the authors added nucleotide-resolution DMS data to the Supplementary figures as this made the data analyzable for we reviewers and the ultimate broad readership. It does not seem that substantial new experiments were performed in responding to reviewers, but rather the authors primarily reanalyzed their data, in particular their DMS probing results, as they describe in their response letter, leading Garcia et al. to nuance their conclusions about Est1 binding vs. stable association with telomerase RNA. Adding a final model should have advanced the impact of the story about what the Pop proteins do in yeast

telomerase. It did clarify what their data do and do not tell us about Pop proteins in telomerase. The authors' goal of the added model (Figure 10) apparently is more to try to explain the breadth of extensive data gathered and reported herein with pop TS alleles at semipermissive temperatures. The model shown needs to be rebuilt, or else at least reframed as an explanation for their observed phenotypes, since Fig 10B begs more questions than it provides answers. Due perhaps to the limitations of the partial, semi-permissive phenotypes employed for pop1 and pop6 TS alleles, the authors' model does not nail down mechanistic understanding of Pop proteins' function in telomerase much beyond Lemieux et al. More impactfully, Garcia et al seem to have made most headway on lessening claims of Lemieux et al., yet this is not the stated impact of this manuscript as written. This manuscript's extensive data overall comprise almost entirely partial and negative results about the Pop1/6 proteins' function(s) in telomerase, as listed by the authors in the rebuttal letter, page 3. Such moderating results are nevertheless of high importance to the field given that they largely contrast with conclusions from the most recent prominent work on the Pop proteins in yeast telomerase, Lemieux et al., since that Cell article concluded that the Pop1/6/7 complex in yeast telomerase RNP were "required" for Est1 and Est2 association with the telomerase RNP, yet Garcia et al. here find that this is far from the case.

More specific concerns that I still have are as follows:

1. The model in Figure 10 shows more total TLC1 RNA in pop cells and these RNA molecules still bind with Est1 and Est2. If the model was accurate, the authors would see greater, rather than less, association between these proteins and TLC1 in co-IP experiments. There seems to be serious disconnect between the data and the model that needs to be resolved by doing more than simply redrawing the model figure.
2. Why can't Est1 (or Est3) and Est2 proteins, which are all at many-fold higher concentrations than the limiting TLC1 RNA as shown by the Zakian group, associate with telomerase RNP in the nucleus in S/G2? Don't the Est proteins have nuclear localization signals and bind to TLC1 directly?
3. The in vivo DMS data for the TLC1 Est1 arm in pop1 vs. pop6 cells are quite different. Is there some explanation why, and can the authors really conclude that the role of Pop1 is the same as Pop6 given the data, such as those in Supp Fig 12 (top row)?
4. The actual data for pop1 vs. pop6 are opposite in Supp Fig 12 compared to in Supp Fig 14. Something must be wrong. This is very confusing when trying to analyze these critically important data for the different pop alleles (which were only first presented in sufficient detail for reviewer analysis in version #2 of this manuscript).
5. The DMS binding data are exhaustive and repeat experiments show very tight reproducibility, which is great. However, they provided quantitative evidence that there is only a very modest reduction of Pop proteins binding to the Est1 arm of TLC1 in pop mutants even at their proposed 7-nt binding site at 30°. This suggests that either Pop6 doesn't actually bind (at least directly) to TLC1 and/or the TS pop alleles are not very penetrant with respect to the relevant binding-competent pool of Pop protein(s). The

authors need to thoroughly explain the very subtle dip in DMS sensitivity in the pop mutant(s) (as for whether it is pop1 or pop6, see points #3 and #4 above, since it seems clearly one and not the other, but there is apparently some problem with the figures).

6. In the authors' stated model in the text, the authors say that Est3 associates as a consequence of Est1 association. Although there is some evidence suggesting this could be true in the literature, it is not a settled matter and the authors have not shown data support his herein, so the authors should moderate their comments so that this is not stated as a matter of fact.

7. The representative microscopy data in Supplementary Figure 19 are useful and important. Looking even at wild-type cells at 24° (A) vs 30° (B), there is a very visually striking increase in TLC1 spots overall, and particularly in the cytoplasm, concomitant with the increase in temperature. What is the explanation for all of the bright TLC1 spots of the representative images in A vs. B, given that this visually evident trend in TLC1 even in wild-type cells is not evident in the quantitative data in Fig 9? Are these microscopy images really representative?

Minor

1. Why is the nucleus, where telomerase primarily resides and ultimately functions, viewed by the authors as an organelle to be avoided due to concern about TLC1 degradation simply because the enzymes that degrade it do so there? It seems going to far to say that telomerase should stay out of the nucleus lest it risk being degraded if it ventures there.

2. The authors state in the title of Supplementary Figure 11 that "TLC1 RNA structure near the Est1 arm is affected in pop cells at 30°C." But the data in Supp Fig 11 do NOT show this region near of the Est1 arm and therefore do not support the conclusion listed for this Supp Figure. Presumably the title should just be changed for Supp Fig 11 to correlate with the region of TLC1 for which it shows results.

Reviewer #2 (Remarks to the Author):

Revised manuscript is easier to read although I still find very long. I feel that writing could be improved but I don't have any objection to the findings.

Minor:

P4L62, extra opening bracket at the citation.

P18L380 position of full stop.

Reviewer #3 (Remarks to the Author):

The revised manuscript and the revision of Garcia et al has addressed my concerns on the reliability of the smFISH protocol. The authors have included several control experiments and raw images. Overall the presented smFISH data is convincing and the conclusion that TLC1 is mislocalized is solid. With regard to the rest of the manuscript, I am not an expert in this field, and I leave the judgement of the novelty and quality of the data to the other reviewers.

January 7, 2020

Dear Reviewers,

We are gratified that reviewers 2 and 3 have no problems with our revised manuscript, except that reviewer 2 would like it to be shortened. Reviewer 3 who is clearly an expert on FISH, says we addressed all of his/her concerns and provide a convincing and solid demonstration that TLC1 is mis-localized in *pop* mutants, one of the most original and unexpected findings of our manuscript.

Reviewer 1's comments are not so positive. For us, these comments are sometimes difficult to understand, often because they are not precise. In some cases, the criticisms reflect an unfamiliarity or inaccurate recollection of the literature (e.g., copy number of telomerase subunits, cytoplasmic assembly of the telomerase holoenzyme) Nonetheless, even this reviewer describes the work as an "important contribution".

Here we do our best to address the points made by reviewer 1. Our comments, which are in italics and underlined, are interspersed with the reviewer's verbatim comments, which are in quotes.

"Ever since the Zakian and Wellinger labs published (in 2015 and 2016) that Pop proteins bind not only to RNase MRP and P RNAs, but also to telomerase RNA, TLC1, the obvious question in the field has been: why do these proteins also associate with telomerase RNA? On a related note, Garcia et al. state that "the goal of this paper is to determine the functional significance of the association of Pop proteins with telomerase." As I stated in my original review, there are a lot of thoroughly presented data in this manuscript. Certainly, these data provide a foundation for studying Pop1 and Pop6 TS alleles and make some headway on understanding what Pop1 and Pop6 do and do not do. But there are limitations to how much has, or can, be learned from partial loss-of-function phenotypes of TS mutants studied at permissive and semi-permissive temperatures." *Of course, there are limits to how much can be learned with any approach, including the approaches used in our paper.*

"Garcia et al. claim that the Pop1 and Pop6 proteins bind to TLC1 to help Est1, and to a lesser extent Est2, stay associated with TLC1, and to help certain pools of telomerase RNP to fully migrate to the nucleus. Comparing the take-home messages with what was published in Lemieux et al. Cell 2016, this Est1/Est2 association conclusion is not novel, even if it was more carefully performed here". *Later in the reviewer's comments, s/he seems to contradict the statement that our findings about Est1/Est2 are not novel.*

(Please see section of reviewer 1's comments that starts out "Lemieux et al. 2016 had concluded that Pop1/6/7 was having a vast, critical influence on telomerase RNP folding and assembly, and Garcia et al. make clear progress on this question"). Our second major finding pointed out by reviewer 1 is not addressed at all in the Lemieux et al paper (i.e., that nuclear localization of TLC1 is Pop-protein dependent).

"And fundamental questions remain unresolved, such as whether Pop proteins have direct roles in catalysis, and other hypotheses proposed by the Wellinger group." No paper can solve all of the fundamental unresolved questions in a field. Moreover, our goal was never to test all of the ideas proposed by Lemieux et al. Indeed, our work was started several years before the publication of Lemieux et al. Surely it is best for a field if important issues are addressed by multiple investigators with different viewpoints and different approaches. "Lemieux et al. 2016 had concluded that Pop1/6/7 was having a vast, critical influence on telomerase RNP folding and assembly, and Garcia et al. make clear progress on this question by showing with *in vivo* DMS probing that only <1% of the analyzed nucleotides in TLC1 were differentially reactive in pop cells compared to 26% in NME1. This demonstrates that Pop proteins only influence TLC1 within the 7-nt position to which they directly bind." Here, the reviewer acknowledges the importance of our work. S/he notes that in contrast to the predictions in Lemieux et al, we show that TLC1 is properly folded when Pop proteins are limiting. Thus, we are able to reject the hypothesis that Pop proteins are needed for Est1/2 binding to TLC1 because they promote the global folding of TLC1. In our opinion, and it seems in reviewer 1's opinion, this is an important contribution. "This sort of progress synthesizes the impact of Garcia et al.'s exhaustive data. By digging deeper into the claims of Lemieux et al., it seems that the Pop1/6/7 complex, as is so often the case with telomerase RNP across species from ciliates to humans, has evolved to coopt a protein complex from a different RNP to promote its own optimal biogenesis." We don't understand the point the reviewer is trying to make here. "This story is not, however, what is stated by Garcia et al. in its currently written form, lessening the impact of the work." As we don't understand the point being made by the reviewer, our paper can't reflect this view. Because of the current Figure 10 model and the overall collection of negative and partial phenotypes, it is questionable as to whether this manuscript warrants publication in a top journal such as Nature Communications."

In our rebuttal of the first reviews, we listed multiple findings in our paper that are not addressed at all in Lemieux et al. In our opinion, not one of these findings is partial and only one is negative but still important (point 3 below). It is inaccurate to describe these findings as negative and/or partial. Here are the findings.

1. Pop proteins are required for maintenance of WT telomere length
2. Pop proteins suppress the copy number of TLC1 (a particularly unexpected result as NME1 and RPR1 levels, the previously known targets of Pop proteins, are reduced in pop cells.)

3. TLC1 is processed normally at both its 5' and 3' ends in pop cells. While a negative result, these data are the first demonstration in any organism of Pop proteins acting in a complex that does not affect RNA processing.
4. TLC1 is mis-folded at the seven nucleotides that are the site of Pop protein binding. Unexpectedly, the rest of TLC1 is properly folded, including at the sites of Est1 and Est2 binding. As pointed out by reviewer 1, this finding eliminates the model proposed in Lemieux et al. We consider this finding to be highly unexpected given that Pop proteins promote the global folding of their other RNA substrates (as hypothesized by others and shown directly in our paper for NME1 RNA).
5. Pop proteins are required for normal levels of telomerase binding to telomeres.
6. By using both DMS-protection and RNA-IP, we show that Est1 and Est2 bind TLC1 at normal levels in pop cells but this binding is unstable. Again, this result is totally unexpected and is a key mechanistic finding of our paper.
7. Telomerase RNA accumulates in the cytoplasm in pop mutants. Again, totally unexpected and one of the key mechanistic findings in our paper.

“But I think that the impact is greater if the authors revisit how they view the impact of their findings by showing how they refine where Lemieux et al. concluded and by modifying Figure 10 (see below).” We feel that we do a good job of explaining how our work differs from Lemieux et al. Reviewer 2 and 3 are satisfied with the revised paper. Without more specifics from this reviewer on what s/he is looking for and with the need to shorten the manuscript, we cannot address this concern.

“In responding to the reviewers, Garcia et al. have rephrased much of the manuscript, reordered some figures, and, importantly, added a final figure showing their model. It was great that the authors added nucleotide-resolution DMS data to the Supplementary figures as this made the data analyzable for we reviewers and the ultimate broad readership. It does not seem that substantial new experiments were performed in responding to reviewers, but rather the authors primarily reanalyzed their data.” There are multiple new experiments in the revised paper. More importantly, we addressed every request for new data by each of the three reviewers.

1. Reviewer 1 asked for only one new experiment, which we provided; i.e., we determined the DNA sequence of the two mutant pop alleles. We present both the gene sequence and the predicted proteins that each encodes in the revised manuscript.
2. As requested by reviewer 2, we document the reduced abundance of Est1 and Est2 at 30°C in mutant versus WT cells in expanded Figure 4 B and D. This reviewer also suggested that we determine mRNA levels of the Pop proteins but seemed okay with our explanation for why mutant pop proteins are less abundant than their WT counterparts. We suspect that showing the altered sequence of Pop proteins helped alleviate this concern.
3. Reviewer 3 asked for several new experiments, all of which were carried out to reviewer 3's satisfaction: (1) we determined the level of background signal with a TLC1

probe in *tlc1Δ* vs WT cells (Supplemental Fig. 17); (2) we present data to support the higher specificity of hairpin probes compared to linear probes (Supplemental Fig. 15); (3) we demonstrate that individual spots in the FISH assay detect individual RNA molecules by two different approaches (Supplemental Figures 18). (4) We present raw FISH image data in WT and pop cells (Supplemental Fig. 19).

“in particular their DMS probing results, as they describe in their response letter, leading Garcia et al. to nuance their conclusions about Est1 binding vs. stable association with telomerase RNA. Adding a final model should have advanced the impact of the story about what the Pop proteins do in yeast telomerase. It did clarify what their data do and do not tell us about Pop proteins in telomerase. The authors’ goal of the added model (Figure 10) apparently is more to try to explain the breadth of extensive data gathered and reported herein with pop TS alleles at semipermissive temperatures. The model shown needs to be rebuilt, or else at least reframed as an explanation for their observed phenotypes, since Fig 10B begs more questions than it provides answers. We don’t understand the reviewer’s points, especially as the reviewer states that Fig 10 clarifies our data. We would need more specifics to understand what the reviewer is asking for. “Due perhaps to the limitations of the partial, semi-permissive phenotypes employed for pop1 and pop6 TS alleles.”

In our opinion, there is no support for the reviewer’s claim that the data are limited or partial because of the use of **semi-permissive** phenotypes. At several points in his/her review, reviewer 1 doesn’t seem to appreciate the rationale for our approach. Each Pop protein is essential for the synthesis of all proteins. Therefore, it is critical to distinguish direct effects of mutant Pop proteins on TLC1 from indirect effects due to an overall decline in protein synthesis. The above comments which twice says that our experiments were done at semi-permissive temperature make us wonder if the reviewer somehow missed that experiments were done at both permissive (24°C) and semi-permissive (30°C) temperatures. As stated clearly in the manuscript, we consider experiments at permissive, not semi-permissive, temperature to be the most valid way to study pop mutants. At 24°C, pop mutations have minimal effects on growth or protein abundance. However, because of their mutant sequence, the abundance of mutant Pop1 and probably mutant Pop6 are lower than their WT counterparts at 24°C. Remarkably, despite the quite modest cellular impact of the pop mutants at 24°C, we see statistically significant effects on TLC1 abundance, the structure of the Pop binding site, its sub-cellular localization, and its stable interaction with Est1/2. The defects in TLC1, which are seen in the absence of defects in growth rate or protein abundance, result in downstream defects in telomere biology that are also statistically significant (e.g., short telomeres). We also detect multiple, statistically significant effects on NME1 RNA, which provides further validation of our approach. Almost all experiments were also done at the semi-permissive temperature of 30°C. Although pop mutants grow more slowly than

WT and show global and specific defects in protein abundance at 30°C, they grow indefinitely at this temperature. Our rationale, as stated clearly in the manuscript, is that phenotypes seen at 24°C and exacerbated at 30°C are likely to be direct effects of mutant Pop proteins on telomerase. Gratifyingly, the TLC1, NME1, and telomere phenotypes are worsened at 30°C. To reiterate, if a phenotype were seen only at 30°C, it could be indirect, given that the abundance of many proteins, including Est proteins, is reduced in pop cells at 30°C. Finally, in each case, we show that the effects of pop mutant on TLC1 and telomeres are reversed by introducing a plasmid-born copy of the POP gene. Because our approach was conservative, it might not have allowed us to see defects in telomerase. However, this conservative approach was robust, as we detected multiple, significant phenotypes at permissive temperature that were worsened at 30°C. We think the reviewer is missing the point by saying that the pop mutants are not penetrant. Doing the experiments at both 24° and 30°C allowed us to rule out indirect effects of Pop protein limitation on general protein synthesis. This approach is a strength, not a weakness. It is remarkable to us that we see so many significant TLC1 and telomere phenotypes when we limit Pop proteins to such a modest extent.

“The authors’ model does not nail down mechanistic understanding of Pop proteins’ function in telomerase much beyond Lemieux et al. More impactfully, Garcia et al seem to have made most headway on lessening claims of Lemieux et al., yet this is not the stated impact of this manuscript as written. This manuscript’s extensive data overall comprise almost entirely partial and negative results about the Pop1/6 proteins’ function(s) in telomerase, as listed by the authors in the rebuttal letter, page 3. Again, as documented above, our results are not partial nor are they mostly negative. Moreover, we provide four novel mechanistic explanations for how Pop proteins affect telomerase that are not shown in Lemieux et al : (1) Pop proteins are required for the stable association of Est1/2 with telomerase RNA; (2) the instability of Est1/2 binding in pop mutants is not due to improper folding of TLC1; (3) When the stability of the holoenzyme is reduced, telomerase RNA accumulates in the cytoplasm, allowing its copy number to rise; (4) Pop proteins have opposite effects on TLC1 and NME1 structure and abundance. These differences suggest to us that it is possible that Pop proteins affect other RNA-protein complexes in a non-catalytic manner. “Such moderating results are nevertheless of high importance to the field given that they largely contrast with conclusions from the most recent prominent work on the Pop proteins in yeast telomerase, Lemieux et al., since that Cell article concluded that the Pop1/6/7 complex in yeast telomerase RNP were “required” for Est1 and Est2 association with the telomerase RNP, yet Garcia et al. here find that this is far from the case. In the end, this reviewer describes our findings as being of high importance. This sounds to us as if reviewer has a positive view of our findings.

“More specific concerns that I still have are as follows: “

1. “The model in Figure 10 shows more total TLC1 RNA in pop cells and these RNA molecules still bind with Est1 and Est2. If the model was accurate, the authors would see greater, rather than less, association between these proteins and TLC1 in co-IP experiments. There seems to be serious disconnect between the data and the model that needs to be resolved by doing more than simply redrawing the model figure.” The reviewer seems to have missed the major and quite novel point that Est1 and Est2 bind at WT levels to TLC1 in pop cells (based on DMS-sensitivity) but this binding is highly unstable (based on RNA-IP). It is our opinion that these points are stated clearly in the end of the discussion and figure 10 legend as well as pictorially in Fig. 10, Reviewers 2 and 3 did not appear to see a serious disconnect between the model and the data. Nonetheless, we emphasized these points even more in the revised manuscript.

2. “Why can’t Est1 (or Est3) and Est2 proteins, which are all at many-fold higher concentrations than the limiting TLC1 RNA as shown by the Zakian group, associate with telomerase RNP in the nucleus in S/G2? Don’t the Est proteins have nuclear localization signals and bind to TLC1 directly?” The paper from the Zakian lab referred to by the reviewer is Tuzon et al, 2011 PLoS Genetics. The points made here by the reviewer reflect a faulty recollection of this paper (see abstract of paper where copy numbers are presented). The published levels for the Est proteins in molecules/cell are: Est1 (71.1±19.2), Est2 (37.2 ±6.5), and Est3 (84.3 ±13.3; as noted in the paper, the Est3 value is probably an overestimate as the epitope tagged Est3 was not fully active). These values were determined in cells grown at 30°C. As shown in the current manuscript, at 24°C, Est1 and Est2 abundance are similar in WT and pop cells. The copy numbers for TLC1 at 30°C are 29 molecules per WT cell (less than 20% variation between replicates; Mozdy and Cech 2006 RNA). As shown in our manuscript, compared to WT cells, TLC1 abundance is twice as high in pop mutants at 24°C (4-6X higher at 30°C); i.e., ~60 molecules per pop cell at 24°C. Thus, there is not a “many-fold” higher concentration of Est1 or Est2 compared to TLC1 in pop cells. Indeed, the data suggest that Est2, the least abundant protein subunit, is present in similar amounts as TLC1 even in WT cells. However, we do not think the ratio of TLC1 to Est proteins is important for the answer to reviewer 1’s question. Rather the critical point is that published data that are well accepted by the field, show that TLC1 associates with its protein subunits in the cytoplasm (see, for example, Wu et al., 2014 Cell Reports: last sentence in summary: “we show that the nuclear export of TLC1 is an essential step for the formation of the functional RNA containing enzyme because blocking TLC1 export....prevents its cytoplasmic maturation and leads to telomere shortening”). To answer the reviewer’s musings, published data show that Est1 has three putative nuclear localization signals, but mutation of these NLSs does not prevent telomerase-mediated telomere lengthening (i.e., Est1 nuclear localization doesn’t require an NLS) (Hawkins and Friedman 2014

Eukaryotic Cell). Also, in response to reviewer 1's comment, we used nls-mapper.iab.keio.ac.jp to see if either Est2 or Est3 had a predicted NLS. Both proteins contain a candidate NLS, but we found no report of their being tested for function. In our opinion, it is not particularly relevant if Est proteins have functional NLS's, the data show (1) the telomerase holoenzyme is assembled in the cytoplasm; (2) cytoplasmic assembly is critical for telomerase action in the nucleus. These conclusions are based on two very different approaches: reduced holoenzyme assembly in estΔ cells (Gallardo et al 2008) and reduced holoenzyme assembly in cells lacking nuclear export factors for TLC1 (Wu et al. 2014). We can now add another requirement for efficient TLC1 re-entry into the nucleus. Stable assembly of the holoenzyme is Pop protein dependent. When Pop proteins are minimally reduced, TLC1 accumulates in the cytoplasm despite WT levels of Est proteins and WT nuclear export factors.

3. "The in vivo DMS data for the TLC1 Est1 arm in pop1 vs. pop6 cells are quite different. Is there some explanation why, and can the authors really conclude that the role of Pop1 is the same as Pop6 given the data, such as those in Supp Fig 12 (top row)?" As noted in both our original and revised manuscripts (lines 108-110), Pop1 and Pop6 do not have identical functions. Pop6 (18.2 kDa) and Pop7 (15.8 kDa) bind as a heterodimer to the P3 regions of NME1 and RPR1 RNAs (Fagerlund et al. 2015). Likewise, Pop6/7 bind to the P3-like region of TLC1 (Lemieux et al, 2016). The Pop6/7 heterodimer stabilizes the binding of the much larger Pop1 (100 kDa), which has a role in recruiting at least some of the other 7-8 proteins in RNase P/MRP (number of proteins is different for the two complexes; as shown in our earlier paper (Lin et al., 2015 Nature Comm), none of the additional subunits are telomerase-associated). Moreover, we show that levels of Pop1 are partially dependent on Pop6. Therefore, we are not surprised that the data are not identical in pop1 and pop6 cells. If anything, we are surprised by how similar the data are in the two mutants over multiple, disparate assays, including the DMS protection data shown in Supplemental fig, 12. It is a tribute to the quality of our methods that our data are sensitive enough to detect a greater effect in pop6 than in pop1 cells in the accessibility of the site of Pop protein binding

4. "The actual data for pop1 vs. pop6 are opposite in Supp Fig 12 compared to in Supp Fig 14. Something must be wrong. This is very confusing when trying to analyze these critically important data for the different pop alleles (which were only first presented in sufficient detail for reviewer analysis in version #2 of this manuscript)." We thank the reviewer for bringing this error to our attention. Indeed, the colors in the line graph of Supp Fig 12 were switched for the pop1 and pop6 alleles. We corrected this mistake.

5. "The DMS binding data are exhaustive and repeat experiments show very tight reproducibility, which is great. However, they provided quantitative evidence that there is only a very modest reduction of Pop proteins binding to the Est1 arm of TLC1 in pop

mutants even at their proposed 7-nt binding site at 30°. This suggests that either Pop6 doesn't actually bind (at least directly) to TLC1 and/or the TS pop alleles are not very penetrant with respect to the relevant binding-competent pool of Pop protein(s). The authors need to thoroughly explain the very subtle dip in DMS sensitivity in the pop mutant(s) (as for whether it is pop1 or pop6, see points #3 and #4 above, since it seems clearly one and not the other, but there is apparently some problem with the figures)."
As with all good science, these issues are answered by the robust statistical analysis that assesses the significance of detected differences in DMS reactivity. The statistical methods developed by the Aviran group and used for the analysis of the DMS data are designed to detect and assess the importance of structural changes, even if not all molecules in the population have the same structure. The rigorous methods for the DMS experiments include a very large number of reads (680,000 reads per A or C) and biological triplicates of DMS treated and untreated controls for each strain. Even though the reviewer views the dip as subtle, the fact that the standard deviations are so tight demonstrates that the data are highly reproducible and statistical analysis shows they are significant. As noted above, it is a tribute to the quality of our methods that our data are sensitive enough to detect a greater effect in pop6 than in pop1 cells in the accessibility of the site of Pop protein binding. It is also gratifying that we detect global unfolding of NME1 RNA with the same methods.

6. In the authors' stated model in the text, the authors say that Est3 associates as a consequence of Est1 association. Although there is some evidence suggesting this could be true in the literature, it is not a settled matter and the authors have not shown data support his herein, so the authors should moderate their comments so that this is not stated as a matter of fact. The data are not shown in our manuscript because they are already published and well accepted by the field (as reflected in review articles). To summarize the literature: (1) Est3 is not recruited to the telomerase holoenzyme in est1Δ cells (Osterhage et al., 2006 NSMB); (2) Est3 does not bind telomeres in est1Δ cells (Tuzon et al. 2011 PLoS Genetics); (3) purified Est1 and Est3 interact directly in vitro (Tuzon et al. 2011 PLoS Genetics). There is an earlier paper from the Lundblad lab that did not detect the Est1-Est3 interaction by a co-IP experiment (Hughes et al., 2000 Curr Biol.). Osterhage used a similar co-IP experiment but their experiments were more sensitive. Typically, positive results, especially when seen in two different labs, trump an earlier negative result.

7. "The representative microscopy data in Supplementary Figure 19 are useful and important. Looking even at wild-type cells at 24 (A) vs 30° (B), there is a very visually striking increase in TLC1 spots overall, and particularly in the cytoplasm, concomitant with the increase in temperature. What is the explanation for all of the bright TLC1 spots of the representative images in A vs. B, given that this visually evident trend in TLC1 even in wild-type cells is not evident in the quantitative data in Fig 9? Are these microscopy images really representative?"

The images are 2D projections of 3D cells. Therefore, the presented visual images should not be used to assess the magnitude of the FISH signal. The size of the signals should be deduced from the presented quantified values. We are pleased that reviewer 3, an expert in FISH, had no concerns with these images.

Minor

1. Why is the nucleus, where telomerase primarily resides and ultimately functions, viewed by the authors as an organelle to be avoided due to concern about TLC1 degradation simply because the enzymes that degrade it do so there? It seems going to far to say that telomerase should stay out of the nucleus lest it risk being degraded if it ventures there. We do not think there is anything in our manuscript to suggest that we view the nucleus as a location that TLC1 is trying to avoid. Rather, we are documenting a heretofore unknown process, the role of Pop proteins in promoting the nuclear re-entry of the telomerase holoenzyme into the nucleus. Here is the point. When TLC1 does not associate with Est1/2, as happens in cells lacking any one of the Est proteins (Gallardo et al 2008) or lacking a nuclear export factor that is needed for TLC1 to leave the nucleus (Wu et al. 2014), telomerase is not active. We show that stable association of Est1/2 with TLC1 requires Pop1/6. Therefore, even at 24°C, when Pop proteins are modestly limited, TLC1 accumulates in the cytoplasm in pop cells. This accumulation occurs even though Est proteins and nuclear export factors are present at WT levels. As the nucleases that degrade TLC1 are nuclear, TLC1 is protected from degradation in pop cells. These data and their interpretation provide a satisfying explanation for the surprising finding that TLC1 abundance is higher in pop cells.

2. The authors state in the title of Supplementary Figure 11 that “TLC1 RNA structure near the Est1 arm is affected in pop cells at 30°C.” But the data in Supp Fig 11 do NOT show this region near of the Est1 arm and therefore do not support the conclusion listed for this Supp Figure. Presumably the title should just be changed for Supp Fig 11 to correlate with the region of TLC1 for which it shows results. The title of the figure legend was correct in the text but not on the figure. we thank the reviewer for pointing out this error.

As requested by reviewer 2, the paper was shortened. The introduction, results, and discussion are now under 6500 words (to be exact, 6498!). The instructions to the authors ask that we highlight changes to the text. The revised manuscript is much shorter than the original. This shortening means that most of the paper has major changes. The order of presentation of experiments is also changed. Therefore, we didn't highlight changes.

Thank you.